# The orphan GPR50 receptor promotes constitutive TGFβ receptor signaling and protects against cancer development

Stefanie Wojciech[1,2,3], Raise Ahmad[1,2,3], Zakia Belaid-Choucair[3,4], Anne-Sophie Journé[1,2,3], Sarah Gallet[5], Julie Dam[1,2,3], Avais Daulat[1,2,3], Delphine Ndiaye-Lobry[1,2,3], Olivier Lahuna[1,2,3], Angeliki Karamitri[1,2,3], Jean-Luc Guillaume[1,2,3], Marcio Do Cruzeiro[1,2,3], François Guillonneau [1,2,3,6], Anastasia Saade[1,2,3], Nathalie Clément[1,2,3], Thomas Courivaud[7], Nawel Kaabi[7], Kenjiro Tadagaki[1,2,3], Philippe Delagrange[8], Vincent Prévot[5], Olivier Hermine[3,4], Céline Prunier[7] & Ralf Jockers[1,2,3]

Transforming growth factor-β (TGFβ) signaling is initiated by the type I, II TGFβ receptor (TβRI/TβRII) complex. Here we report the formation of an alternative complex between TβRI and the orphan GPR50, belonging to the G protein-coupled receptor super-family. The interaction of GPR50 with TβRI induces spontaneous TβRI-dependent Smad and non-Smad signaling by stabilizing the active TβRI conformation and competing for the binding of the negative regulator FKBP12 to TβRI. GPR50 overexpression in MDA-MB-231 cells mimics the anti-proliferative effect of TβRI and decreases tumor growth in a xenograft mouse model. Inversely, targeted deletion of GPR50 in the MMTV/Neu spontaneous mammary cancer model shows decreased survival after tumor onset and increased tumor growth. Low GPR50 expression is associated with poor survival prognosis in human breast cancer irrespective of the breast cancer subtype. This describes a previously unappreciated spontaneous TGFβ-independent activation mode of TβRI and identifies GPR50 as a TβRI co-receptor with potential impact on cancer development.

[1] Inserm, U1016, Institut Cochin, Paris 75014, France. [2] CNRS UMR 8104, Paris 75014, France. [3] Université Paris Descartes Université Sorbonne Paris Cité, Paris 75006, France. [4] Hôpital Necker, CNRS UMR 8147, Paris 75015, France. [5] Jean-Pierre Aubert Research Center, U837 Lille, France. [6] Plateforme Proteomique 3P5 de l'Université Paris Descartes, Paris 75014, France. [7] Sorbonne Université, UPMC Univ Paris 06, INSERM Centre de Recherche Saint-Antoine (CRSA), 75012 Paris, France. [8] Institut de Recherches SERVIER, Suresnes 92150, France. These authors contributed equally: Stefanie Wojciech, Raise Ahmad. Correspondence and requests for materials should be addressed to R.J. (email: ralf.jockers@inserm.fr)

Transforming growth factor β (TGFβ) is a cytokine, which regulates many cellular processes and plays an important role in normal embryogenesis and tissue homeostasis due to its effects on proliferation, differentiation, or apoptosis[1–4]. TGFβ elicits its effects through two single-transmembrane (TM) spanning serine/threonine (Ser/Thr) kinases called type I and type II TGFβ receptors (TβRI and TβRII, respectively)[5]. Binding of TGFβ to TβRII triggers the recruitment of TβRI[6]. The constitutively active TβRII kinase activates TβRI by phosphorylating several Ser/Thr residues in the highly conserved GS region ([185]TTSGSGSG[192]) located N-terminal to the kinase domain of TβRI[7]. This induces the so-called "inhibitor to substrate" activatory switch, which consists in the dissociation of the FKBP12 inhibitor and the subsequent binding of Smad2/3 proteins[8]. Phosphorylation of Smad2/3 by the TβRI kinase[9] induces their dissociation from the receptor, which then form a dimeric complex with the Co-Smad, Smad4, translocate to the nucleus, and regulate gene transcription upon DNA binding[10].

Over the last two decades several regulators have been identified that allow a context-dependent integration of the core signaling pathway[2]. Among those, SARA potentiates Smad recruitment to TβRI[11], Smad7 competes with Smad2/3 for TβRI binding[12], and TMEPAI interferes with Smad2/3 phosphorylation[13].

Because many cancers of epithelia develop resistance to the negative growth-regulatory effects of TGFβ, it has been postulated that one of the mechanisms whereby cells undergo neoplastic transformation and escape from normal growth control involves an altered response to TGFβ. Cancer cells can acquire resistance to the antiproliferative effect of TGFβ by a number of different mechanisms, including defects in TGFβ cell surface receptors and mutational inactivation of downstream effector components of the signaling pathways, including Smad proteins. For example, TβRII and Smad4 mutations were found in a variety of human tumors[3,4].

G protein-coupled receptors (GPCRs), also called 7TM spanning proteins, represent the most abundant class of cell surface receptors with ~800 members. GPCRs are major drug targets accounting for up to 30% of currently marketed drugs[14]. GPCRs have the potential to interact with themselves (homomers) or with other receptors (heteromers)[15]. Within these heteromeric complexes, allosteric regulation of one protomer by the other is often observed. Approximately 100 GPCRs are still orphans for which no endogenous ligand has been identified so far. Ligand-independent functions are more and more reported for orphan GPCRs[16,17]. This includes the allosteric regulation of the function of other GPCRs in heteromeric protein complexes.

GPR50 is an orphan GPCR, which shares highest sequence homology with melatonin receptors[18,19]. The large carboxyl terminal tail (C-tail) of GPR50 functions as scaffold for interacting partners[20,21] and modulates the activity of other membrane receptors such as the melatonin $MT_1$ receptor within heteromeric complexes[22]. A frequent sequence variant (minor allelic frequency = 0.4) that lacks four amino acids, [502]TTGH[505] (GPR50Δ4) is associated with mental disorders[23] and altered lipid metabolism[24].

We report here the formation of a new molecular complex between TβRI and the orphan GPR50 that does not require TβRII. GPR50 enhances the basal, TGFβ-independent, capacity of TβRI to activate Smad2/3 (and non-canonical pathways), most likely by prohibiting binding of the inhibitory FKBP12 to TβRI and by stabilizing the active TβRI conformation in early endosomes. Ectopic expression of GPR50 protects against tumor development and its absence is pro-tumorigenic in animal models. Low GPR50 levels are associated with poor survival prognosis in human breast cancer (independently of the breast cancer subtype).

## Results

**GPR50 interacts with TβRI**. When searching for GPR50-interacting partners by tandem affinity purification coupled to mass spectrometry, we identified five unique peptides corresponding to the TβRI in HEK293T cells expressing the frequent GPR50Δ4 variant but not in naive HEK293T cells (Fig. 1a). Highest expression of GPR50 is observed in tanycytes lining the third ventricle at the level of the hypothalamus and several peripheral tissues[25,26]. Colocalization between GPR50 and TβRI was observed in mouse brain slices of tanycytes, tanycyte primary cultures, human lung carcinoma NCI-H520 cells expressing endogenous GPR50, and in transfected HeLa cells as revealed by immunofluorescence microscopy and proximity ligation assay (PLA, Fig. 1b–d; Supplementary Fig. 1a-c). Co-immunoprecipitation (coIP) experiments in tanycytes, NCI-H520 cells, MDA-MB-231 cells, 4T1 cells and in the mouse cortex and lungs of C57/Bl6 mice, but not of GPR50ko mice, confirmed the interaction (Fig. 1e–h; Supplementary Fig. 1e, 5c) (see also TβRI/TβRII coIP in NCI-H520 cells as positive control; Supplementary Fig. 1d). GPR50 formed a molecular complex with TβRI, likely composed of at least two subunits of each receptor (like the TβRI/TβRII complex), as revealed by coIP of HEK293T cell lysates treated with the DSS cross-linker (Supplementary Fig. 1f). The TβRI/GPR50 complex was formed to a similar extent for the TβRI wt and constitutively active (ca) T204D mutant form as well as upon TGFβ stimulation (Supplementary Fig. 1g). The TM domain of GPR50 was necessary for the interaction (Supplementary Fig. 1h). The interaction was further confirmed in the bioluminescence resonance energy transfer (BRET) donor saturation assay in intact HEK293T cells coexpressing a fixed amount of the TβRI-*Renilla* luciferase 8 (Rluc8) fusion protein and increasing amounts of TβRI-yellow fluorescent protein (YFP), TβRII-YFP, GPR50wt-YFP, or GPR50Δ4-YFP fusion proteins (Fig. 1i). Hyperbolic saturation curves reflected specific interactions between TβRI homomers as expected, but also between TβRI/GPR50wt heteromers and TβRI/GPR50Δ4 heteromers ($BRET_{50} = 1.342 \pm 0.185$, $0.025 \pm 0.006$, and $0.031 \pm 0.010$, respectively; $n = 3$–4) (Fig. 1i, left graph). TβRI/TβRII complexes were also formed at a similar potency to TβRI/GPR50 heteromers ($BRET_{50} = 0.046 \pm 0.008$; $n = 3$). Consistent with the known ligand-promoted TβRI/TβRII complex formation, $BRET_{50}$ values were decreased ($BRET_{50} = 0.0059 \pm 0.0021$; $n = 3$) upon TGFβ stimulation (Fig. 1i, right graph), while BRET signals between TβRI and GPR50 were insensitive to TGFβ treatment (Supplementary Fig. 1i, j).

The expressions of the proteins interacting with TßRI are frequently upregulated by TGFß forming a regulatory loop. 24-hour stimulation of NCI-H520 cells with TGFß increased the amount of GPR50 mRNA and protein (Fig. 1j). Taken together, our data demonstrate the existence of a new molecular complex between TβRI and GPR50 in various tissues. Whereas the activation state of TβRI is independent of TGFß in the complex, the expression of GPR50 can be upregulated by TGFß.

**GPR50 induces canonical and non-canonical TβRI signaling**. We then studied the effect of GPR50 expression on the canonical Smad2/3 signaling pathway. Ectopic expression of the two human GPR50 forms in HEK293T cells substantially increased the basal Smad2 and Smad3 phosphorylation (p-Smad2 and p-Smad3) as compared to TGFβ stimulation and was blocked by the TβRI kinase-specific SB431542 inhibitor (Fig. 2a; Supplementary Fig. 2a, b). Consistent with a tonic effect of GPR50 on Smad activation, silencing of GPR50 in NCI-H520 cells decreased p-Smad2 levels to the same extent as TβRI silencing whereas TGFβ treatment increased it (Fig. 2b; Supplementary Fig. 2c). Similarly,

decreased basal p-Smad2 and p-Smad3 levels were observed in the hypothalamus and cortex of GPR50ko mice as compared to wt mice (Fig. 2c, d). Phosphorylation levels of other signaling molecules like JAK2 and STAT3 were unchanged demonstrating the specificity for Smad2/3 (Fig. 2c). To gain insights in the mechanism underlying this spontaneous Smad phosphorylation

in the presence of GPR50, we studied the subcellular localization of TβRI. Whereas TβRI did not colocalize with the early endosomal EEA1 marker proteins, substantial colocalization was observed when the wt or Δ4 form of GPR50 was co-expressed or when cells were stimulated with TGFβ (Fig. 2e). Sustained activation at an amplitude similar to that observed upon TGFβ

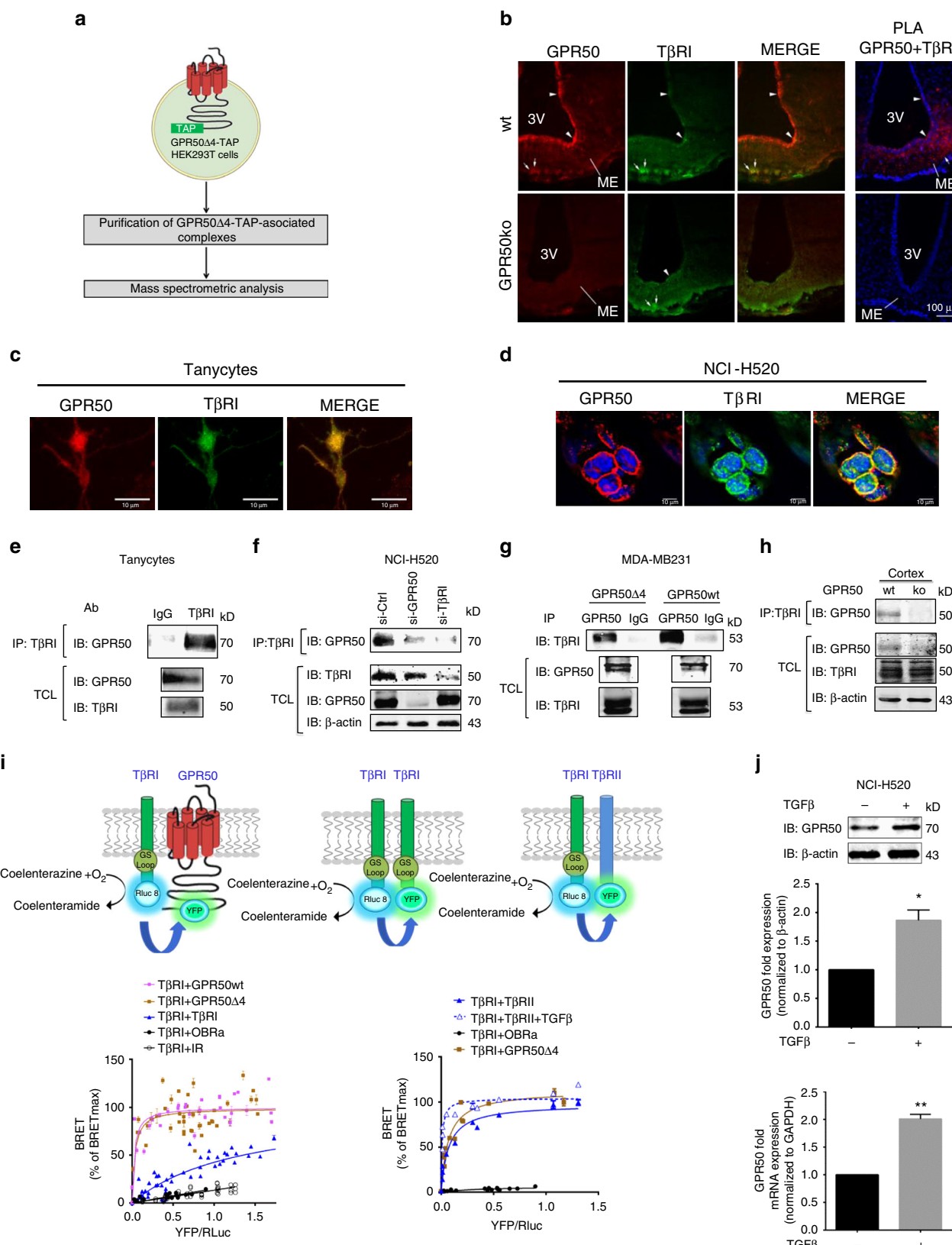

stimulation was also observed at further downstream levels in HEK293T cells (i) complex formation between Smad2/3 and Smad4 (Supplementary Fig. 2d, e), (ii) nuclear translocation of Smad2/3 (Fig. 2f), (iii) ARE-dependent (Fig. 2g) and CAGA-dependent reporter gene activation (Supplementary Fig. 2f), and the expression of the TGFβ target gene Snail (Fig. 2h). Expression of GPR50 in HEK293 cells also promoted spontaneous activation of non-canonical TβRI pathways such as p38 (Fig. 2i; Supplementary Fig. 5d). Expression of the TM domain of GPR50 (GPR50TM, without C-tail) or the cytosolic C-tail (GPR50Cter) alone or fused to the 7TM domain of the melatonin MT$_2$ receptor (MT$_2$-Cter GPR50) were unable to replicate the spontaneous activation of the Smad pathway indicating that the structural integrity of GPR50 is necessary for the effect (Supplementary Fig. 2g-i). Canonical GPCR-associated G protein (Gi, Gq) or β-arrestin activation seems not to be involved as inhibition of these pathways did not abolish the constitutive GPR50-dependent increase in p-Smad2 levels (Supplementary Fig. 2j). Collectively, expression of full-length GPR50 (wt and Δ4 variant) promotes spontaneous and specific activation of TβRI-dependent canonical (Smad2/3) and non-canonical signaling in vitro and in vivo most likely by activating TβRI.

**GPR50 interferes with FKBP12 binding to TβRI.** To further explore the molecular mechanism by which GPR50 potentiates the basal activation of the TβRI/Smad pathway, we determined whether expression of GPR50 in HEK293 cells induces TGFβ production or secretion. This was not the case as treating naive cells with supernatants of cells expressing GPR50 did not phosphorylate Smad (Supplementary Fig. 3a). Any impact of GPR50 on TGFβ affinity (Supplementary Fig. 3b) was also excluded by performing $^{125}$I-TGFβ-binding assay in cells overexpressing GPR50. We also ruled out the possibility that GPR50 scaffold Smad2 or Smad4 by western blot detection of GPR50 following immunoprecipitation of Smad (Supplementary Fig. 3c). Finally, we considered the involvement of the immunophilin FK506-binding protein (FKBP12) which binds and stabilizes TβRI in its inactive conformation thus preventing spontaneous TβRI activation[8,27]. Here we show that the interaction of TβRI and FKBP12 is diminished by 50% upon TGFβ stimulation of HEK293T cells and by 50–60% in cells expressing GPR50 (Fig. 3a). Consistently, the amount of TβRI co-immunoprecipitated with FKBP12 was largely increased in mouse brain and lung lysates prepared from GPR50ko mice as compared to C57/Bl6 wt mice (Fig. 3b, c). Competition of FKBP12 and GPR50 for TβRI binding was further confirmed by BRET showing a 40% reduction as compared to a 20% reduction of the BRET signal of the TGFβ-activated TβRI/TβRII

couple in the presence of FKBP12 (Fig. 3d; Supplementary Fig. 3d). Similarly, the expression of GPR50Δ4 was able to interfere with FKBP12 binding to TβRI conversely to the Cter or the TM domain of GPR50 alone (Supplementary Fig. 3e). The FK506 macrolide compound, known to bind to FKBP12 at a site that competes with binding to TβRI[28] increased basal Smad3 phosphorylation, which was not further increased by GPR50 (Fig. 3e). Overexpression of FKBP12 abolished basal Smad3 phosphorylation irrespective of the expression of GPR50 (Fig. 3e). Collectively, these results further confirm a common action mechanism where FKBP12 and GPR50 compete for binding to TβRI.

To elucidate the molecular basis of this competition, we compared the sequence of the C-tail of GPR50 and FKBP12 and revealed a repetitive 5 amino acid motif (**AXZHP**) (X = Ala, Thr, Ser; Z = Gly, Ser) in GPR50 similar to the $^{84}$ATGHP$^{88}$ motif of FKBP12 (Fig. 3f, upper part). Interestingly, the $^{84}$ATGHP$^{88}$ motif of FKBP12 corresponds to a loop that is part of the binding pocket in the co-crystal structure of FKBP12 and the unphosphorylated GS region of the kinase domain of TβRI[29] (Fig. 3f, lower part). Mutants of the ATGHP loop of FKBP12 (FKBP12-H87L and FKBP12ΔHP) were indeed unable to bind to TβRI (Supplementary Fig. 3f) similar to the previously reported TβRI-P194K mutant[27]. Inversely, disruption of the $^{495}$ATSHP$^{499}$ motifs in GPR50 (H498L and ΔHP mutants) located next to the $^{502}$TTGH$^{505}$ deletion in the Δ4 variant, fully restored FKBP12 binding to TβRI to levels seen in the absence of GPR50 (Fig. 3g). These observations provide a molecular basis explaining the competition of GPR50 and FKBP12 for binding to TβRI.

Taken together, displacement of FKBP12 together with the stabilization of the active conformation of TβRI by GPR50 are likely to contribute to the spontaneous stimulatory effect of GPR50 on TβRI signaling.

**GPR50 activates TβRI signaling in the absence of TβRII.** According to the current dogma, TβRII fulfills two essential functions in the TβR activation process, namely binding of TGFβ and transphosphorylation of TβRI in the GS region[7]. To clarify the role of TβRII in the context of the TGFβ-insensitive GPR50/TβRI complex, we used gastric carcinoma SNU638 cells, which do not respond to TGFβ as they are devoid of TβRII expression[30] (Supplementary Fig. 4a). In agreement with previous reports exogenous expression of TβRII was sufficient to restore TGFβ responsiveness[31] (Fig. 4a). Expression of GPR50 promoted the formation of the GPR50/TβRI complex (Fig. 4b) and increased basal pSmad3 levels (Fig. 4a) in a TβRI kinase activity-dependent manner (Fig. 4a). No further increase was observed upon TGFβ addition (Supplementary Fig. 4b).

**Fig. 1** GPR50 interacts with TβRI and its expression is upregulated by TGFβ. **a** Tandem affinity purification of naive HEK293T cells stably expressing GPR50Δ4-TAP. After purification, mass spectrometry was employed for protein identification. **b** Left panel shows confocal images of GPR50 and TβRI staining in the lining of the third ventricle of brain slices of wt (top) and GPR50ko mice (bottom). Right panel visualizes TβRI/GPR50 interaction by proximity ligation assay (PLA) in the median eminence (ME) and third ventricle (3 V) of wt (top) and GPR50ko (bottom) mice (scale: 100 µm). White arrows depict immunoreactive (IR) regions. See also Supplementary Fig. 1a. **c** Confocal images of GPR50 (red) and TβRI (green) staining in primary rat tanycytes (scale: 10 µm). **d** Colocalization of GPR50 (red) and TβRI (green) in NCI-H520 cells. (scale: 10 µm). **e** Co-immunoprecipitation of GRP50 and TβRI in lysates of primary rat tanycyte cultures. Lysates with IgG served as negative control. **f** Co-immunoprecipitation of GRP50 and TβRI in the lysates of NCI-H520 after silencing either GPR50 (si-GPR50) or TβRI (si-TβRI). Control si-RNA (si-Ctrl) served as control. **g, h** Co-immunoprecipitation of GRP50 and TβRI in lysates of MDA-MB231 cells (**g**) and cortex (**h**) isolated from wild type (wt) or GPR50ko mice. IgG served as negative control. **i** Upper part depicts schematic representation of BRET assay to study the interaction between TβRI-Rluc8 and GPR50-YFP or TβRI-YFP (left and middle scheme) and right scheme between TβRI-Rluc8 and TβRII-YFP. Lower part shows BRET donor saturation curves in HEK293T cells (left: constant expression level of TβRI-Rluc8 and increasing levels of TβRI-YFP, GPR50Δ4-YFP or GPR50wt-YFP; right: constant expression level of TβRI-Rluc8 and increasing levels of GPR50Δ4-YFP or TβRII-YFP with TGFβ stimulation (0.6 nM, 30 min at 37 °C)). IR-YFP and OBRa-YFP served as negative control. BRET signals were normalized to BRET$_{max}$ values. Curves are obtained from three independent experiments performed in triplicates. **j** NCI-H520 cells were starved and stimulated for 24 h with TGFβ (2 ng/mL). GPR50 expression was checked by Immunoblotting and Q-PCR. (Mean ± s.e.m., $n = 3$ independent experiments, *$p < 0.05$; **$p < 0.01$, two-tailed unpaired Student's $t$-test). Representative results are shown for **e**–**h** and **i**. See also Supplementary Fig. 1

To explore the capacity of GPR50 to directly activate TβRI in the TβRI/GPR50 complex we incubated purified TβRI with precipitated GPR50 and observed a 1.6-fold increase in the TβRI kinase activity in an in vitro kinase assay (Supplementary Fig 4c). TGFβ treatment of SNU638 cells expressing TβRII-induced phosphorylation of TβRI at Ser-165 to a similar extent as GPR50Δ4 and GPR50wt expression (Fig. 4c). The effect of GPR50

was inhibited by the TβRI kinase-specific SB431542 inhibitor suggesting that GPR50 allosterically regulates the TβRI kinase leading to TβRI auto-phosphorylation at Ser-165 (Supplementary Fig. 4d).

Overexpression of FKBP12 in SNU638 cells reduced the basal and TGFβ-induced Smad3 phosphorylation in the presence of TβRII in accordance with previous reports (Fig. 4d). Similar

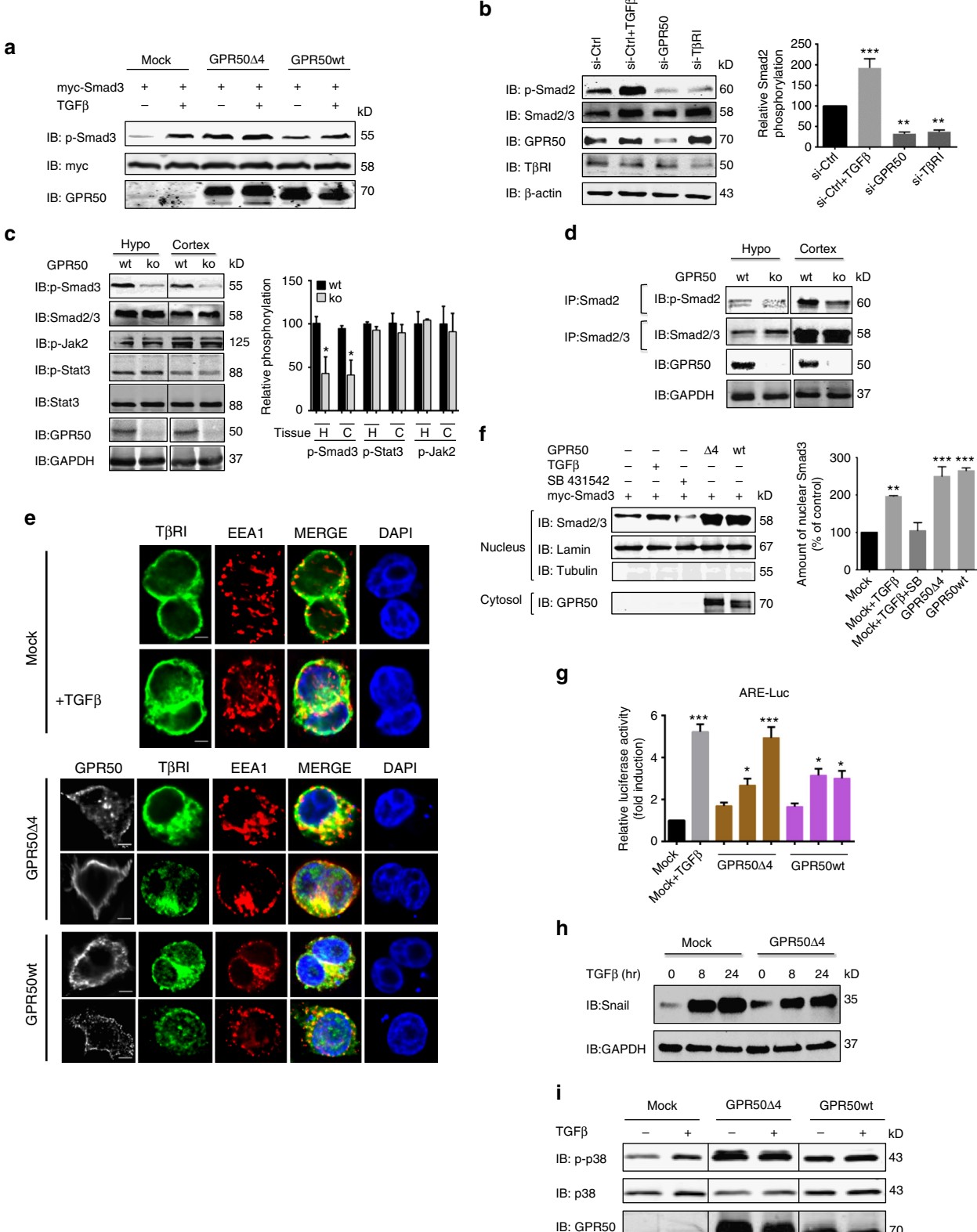

observations were made in cells expressing GPR50 further consolidating the hypothesis that GPR50 competes with FKBP12 for binding to TβRI (Fig. 4d). Taken together, these results show that the formation of the GPR50/TβRI complex does not require TβRII. Similar to TβRII, GPR50 activates the TβRI kinase activity through TβRI phosphorylation and promotes Smad signaling in a manner that can be inhibited by FKBP12 overexpression.

**GPR50 expression mimics TGFβ-mediated cellular responses.** To study the relevance of the GPR50/TβRI complex on cell proliferation and tumor development we choose the MDA-MB-231 breast cancer cell line in which we were able to recapitulate the spontaneous Smad2 and Smad3 phosphorylation in the presence of GPR50 (Supplementary Fig. 5a). The effect of GPR50 on cell proliferation was studied in MDA-MB-231 cell pools stably expressing similar levels of GPR50wt or GPR50Δ4 (Fig. 5a). TGFβ treatment of the mock transfected cell pool significantly (*$p < 0.05$, **$p < 0.01$,***$p < 0.001$ one-way ANOVA with Tukey multiple comparison post hoc test) diminished cell proliferation as expected and silencing of TβRI abolished this effect (Fig. 5b). Even in the absence of TGFβ, expression of GPR50wt or GPR50Δ4 alone decreased cell proliferation to the same extent, while the blockage of the proliferation was reversed by TβRI silencing indicating that GPR50, similar to TGFβ, inhibits the proliferation of MDA-MB-231 cells through TβRI (Fig. 5b). Similarly, silencing of GPR50 in NCI-H520 cells significantly (*$p < 0.05$, **$p < 0.01$, one-way ANOVA with Dunnett's post hoc test) increased their proliferation rate, copying the effect of TβRI silencing in these cells (Fig. 5c; Supplementary Fig. 5b). This indicates that GPR50 affects cell proliferation in breast cancer and lung carcinoma-derived cells.

The 4T1 mammary carcinoma cell line that is resistant to the growth inhibitory effect of TGFβ[32] was chosen to specifically study the role of GPR50 in cellular migration. In the presence of GPR50 a TβRI/GPR50 complex was also formed in 4T1 cells (Supplementary Fig. 5c). Similar to TGFβ, expression of GPR50 promoted 4T1 cell migration (Fig. 5d) indicating that GPR50 exhibits also TGFβ-like properties in respect to cell migration. Consistently, this was accompanied by increased Smad3 phosphorylation and activation of several non-canonical TGFβ signaling pathways such as p38 and AKT and to a lesser extent ERK1/2 (Supplementary Fig. 5d).

We then tested the capacity of GPR50 to modulate the anchorage-independent growth. We observed in a colony formation assay that the number of colonies formed during 3 weeks was diminished in mock MDA-MB-231 cell pools treated with TGFβ and in cell pools expressing GPR50wt or GPR50Δ4 in the absence of TGFβ (Fig. 5e). The effect of GPR50

overexpression on the tumor-promoting capacity of MDA-MB-231 cells was then studied in xenograft experiments. The volume of tumors developed in nude mice injected in the flanks with GPR50wt or GPR50Δ4 expressing MDA-MB-231 cell pools was significantly (*$p < 0.05$, two-way ANOVA with unpaired $t$-test) diminished compared to mice injected with mock cells starting from day 23 until the end (day 34) (Fig. 5f). Collectively, these data show that GPR50 expression mimics TGFβ-mediated cellular responses.

**GPR50 protects against tumor development in MMTV/Neu mice.** To more directly address the role of endogenous GPR50 levels on tumor growth, the absence of GPR50 in vivo was studied in GPR50 knockout (GPR50ko) mice that were generated in the MMTV/Neu background. Expression of the Neu receptor tyrosine kinase, the mouse ortholog of the ErbB-2 receptor in the mammary epithelium of these transgenic mice, leads to the spontaneous development of mammary tumors[33]. This model is particularly interesting as a constitutively active TβRI, carrying several mutations preventing FKBP12 binding and abolishing receptor kinase activity, has been shown to impair Neu-induced primary breast tumor formation[34]. Phenotypic characterization of GPR50ko/MMTV/Neu mice showed a tendency for decreased survival ($p = 0.056$; Kaplan–Meier survival curve with log-rank Mantel–Cox test; Supplementary Fig. 6a), no impact on tumor onset (Supplementary Fig. 6d) but a drastic reduction of overall survival after tumor onset from 70 to 28 days was observed (Fig. 6a, b). The number of tumors, their size and weight were increased (Fig. 6c, d; Supplementary Fig. 6b, c). Collectively, these results show that endogenous GPR50 has anti-proliferative effects, like TGFβ, and has protective effects against tumor development in the MMTV/Neu model.

**Low GPR50 expression and poor survival prognosis in breast cancer.** Analysis of cancer vs. normal tissue with the Oncomine 3.0 database revealed no association with GPR50 overexpression but a significant association with GPR50 underexpression in 10 studies including a fivefold decrease in invasive lobular breast carcinoma ($p < 0.05$) (Fig. 6e; Supplementary Tables 1-6).

To further verify the prognostic significance of GPR50 in breast cancer, we took advantage of a larger breast cancer survival database (TCGA, GEO, and EGA; http://kmplot.com/breast/). Kaplan–Meier analysis confirmed that low GPR50 expression is associated with poor relapse-free survival prognosis in 3951 breast cancer patients (Fig. 6f). Interestingly, the survival of the patients decreased with low expression of GPR50 even after accounting for the four molecular breast cancer subtypes: luminal A, luminal B, Her-2, and basal-like (Fig. 6g). Similar data were

**Fig. 2** GPR50 promotes ligand-independent activation of TβRI signaling. **a** HEK293T cells expressing myc-Smad3, and GPR50Δ4 or GPR50wt were starved overnight and stimulated with TGFβ (2 ng/mL; 1h). Smad3 phosphorylation was checked. Similar results were obtained in at least two additional experiments. **b** p-Smad2 detection in NCI-H520 cells following the silencing of GPR50 and TβRI Control si-RNA (si-Ctrl) with and without TGFβ stimulation (2 ng/mL; 1h) served as control. Densitometric analysis of three independent experiments (Mean ± s.e.m., $n = 3$ independent experiments, ***$p < 0,001$, one-way ANOVA with Dunnett's post hoc test). **c, d** Detection of p-Smad3, p-Stat-3, p-Jak2 (**c**) and p-Smad2 (**d**) in lysates of hypothalamus and cortex of wt and GPR50ko mice. p-Smad2 was detected after precipitation of total Smad2/3. Quantification is shown on the right side of panel c. H hypothalamus, C cortex; Densitometric analysis of three independent experiments (Mean ± s.e.m., $n = 3$ independent experiments, **$p < 0.01$, two-tailed unpaired $t$-test). **e** Confocal images of HeLa cells expression GPR50 and TβRI alone or together showing TβRI colocalization with early endosome marker (EEA1) with TGFβ stimulation (scale: 10 μm). **f** (Left panel) Nuclear extracts of HEK293T cells treated with TGFβ (2 ng/mL, 1 h) and SB431542 (10 μM; O/N) and expressing indicated proteins. (Right) Densitometric analysis of three independent experiments (Mean ± s.e.m., $n = 3$ independent experiments, *$p < 0.05$, **$p < 0.01$, one-way ANOVA with Dunnett's post hoc test). **g** HeLa cells were transfected with a a *Firefly*-Luciferase-coupled ARE- (together with FAST-2) reporter gene construct and *Renilla* Luciferase for normalization. The cells were transfected with empty (Mock± TGFβ; 0.5 ng/mL, 8 h) or 10, 50 and 100 ng of GPR50Δ4 and GPR50wt constructs (Mean ± s.e.m., $n = 3$ independent experiments, *$p < 0.05$, ***$p < 0.001$ one-way ANOVA with Dunnett's post hoc test). **h** 4T1 cells stably expressing either empty plasmid (Mock) or GPR50Δ4 were stimulated with TGFβ (2 ng/mL; 0, 8, 24 h). Snail expression was analyzed by immunoblotting. **i** HEK293T cells expressing indicated plasmids as in (**a**) and stimulated with TGFβ (2ng/mL, 1 h) to reveal p-p38 protein. Representative results are shown for **a**, **d**, **e**, **h**, **i**. Similar results were obtained in at least two additional experiments. See also Supplementary Fig. 2

obtained from Prognoscan.org[35] (Supplementary Fig. 6e-h). Taken together, low GPR50 expression appears to be an independent marker of poor survival prognosis in breast cancer.

## Discussion

We describe here a previously unappreciated activation mode of TβRI when engaged into a molecular complex with the orphan GPR50 receptor. Whereas in the classical mode of action, binding of TGFβ to TβRII promotes the association and phosphorylation of TβRI by TβRII, the physical interaction of GPR50 with TβRI leads to the spontaneous and ligand-independent phosphorylation and activation of TβRI and induction of canonical (Smad) and non-canonical downstream signaling (Fig. 7). Spontaneous activation of TβRI goes along with reduced binding of FKBP12 to TβRI and is sufficient to promote TGFβ-like responses like inhibition of proliferation, promotion of migration, reduction of anchorage-independent growth in vitro. In vivo GPR50 protects against tumor development in mice and low GPR50 levels associate with poor survival prognosis in human breast cancer.

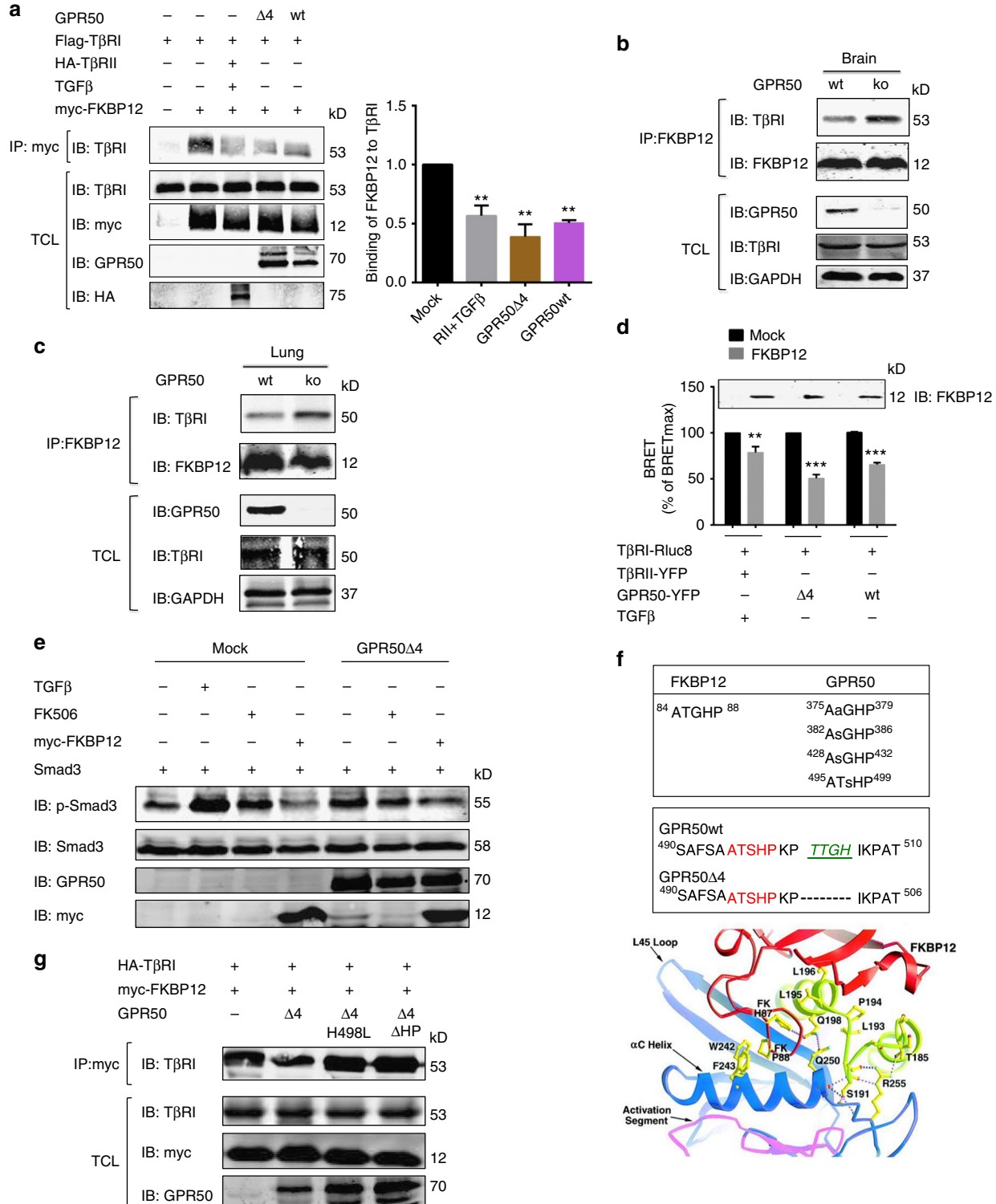

Complex formation between GPR50 and TβRI is likely to fine-tune the TβRI signaling capacity in a cell context-dependent manner, it represents the first example of direct regulation between a member of the GPCR super-family and TβRs and provides further support for the physiological relevance of the concept of ligand-independent functions of orphan GPCRs.

The GPR50/TβRI complex is the first example of a previously unrecognized crosstalk between TGFβ receptors and GPCRs at the receptor level. The capacity of GPCRs to engage into molecular complexes with other receptors, either of the same family (GPCR heteromers) or with proteins of other receptor families or transporters is increasingly recognized[36,37]. Indeed, such complexes significantly diversify the repertoire of pharmacological targets with a limited number of proteins. Such complexes might be of particular importance for orphan GPCRs. There exists indeed more than 100 orphan GPCRs for which no ligand has been identified yet. Apart from the ligand-dependent function that still has to be elucidated, an alternative hypotheses based on the existence of ligand-independent functions of orphan GPCRs is emerging[16,17]. This also applies to GPR50, which has been shown to heteromerize with the melatonin $MT_1$ receptor and to inhibit ligand binding, G protein coupling and β-arrestin recruitment to $MT_1$ in the common $GPR50/MT_1$ complex[22] and the Nogo/GPR50 complex[38]. As in the GPR50/TβRI complex, the C-terminal domain of GPR50 appears to play an important role in the modulation of the function of the interacting partner. The GPR50 C-tail domain most likely has to be in close proximity to TβRI, i.e., in a molecular complex, a condition that cannot be met when the C-tail is expressed as a soluble protein nor when it is fused to the $MT_2$ receptor that does not interact with TβRI. TβRI is one of seven-type I receptor family members. Whether GPR50 also regulates the function of the other members, four activin-like receptors and two BMP receptors, will be interesting to study as all members display strong sequence homology, underlie the same activation modus and bind to FKBP12[28].

Constitutive TβRI activity has been previously observed for receptors of the TGFβ family. The TβRI-T204D mutant, which does not interact with TβRII anymore, constitutively activates the Smad2/3 signaling pathway. Position 204 is part of the RTI sequence adjacent to the kinase domain, which is not phosphorylated itself but has a positive allosteric effect on the phosphorylation of the GS sequence. This mutant receptor does not interact anymore with FKBP12 and shows increased TβRI kinase activity in vitro[7]. This mutant shows that, similar to GPR50, activation of TβRI is possible in the absence of TβRII, likely by stabilizing an active conformation of TβRI. However, in contrast to the GPR50/TβRI complex, the TβRI-T204D mutant is still sensitive to TβRII as TGFβ stimulation generates a further increase of Smad2/3 signaling.

Another reported case of a constitutively active TβRI is the naturally occurring R206H mutant of the activin A receptor type I (ACVRI). This mutant is associated with *fibrodysplasia ossificans progressiva*, a rare genetic and catastrophic disorder characterized by progressive heterotopic ossification[39]. Similar to the constitutively active T204D mutant of TβRI, the R206H mutation is located in the part of the GS region that is close to the kinase domain and that allosterically regulates phosphorylation of the GS sequence. Molecular analysis revealed modest constitutive activity and impaired FKBP12 binding[40]. Recent studies indicate that the simple presence of TβRII, but not its kinase activity nor TGFβ-binding capacity, is necessary for the constitutive activity of the R206H mutant. This suggests that in the context of an activating TβRI mutant, scaffolding function of a co-receptor like TβRII is sufficient for TβRI signaling[41].

More evidence for constitutive TβRII-independent activation of TβRI comes from DAF-1, the TβRI of *C. elegans*[42]. Interestingly, signaling of DAF-1 can occur in the absence of TβRII (DAF-4) kinase activity and promotes larval development. Differences in the structure of the GS region of DAF-1 in comparison to other TβRI isoforms are possibly at the origin of this autonomous signaling capacity of TβRI. In addition, DAF-1 can also signal through the more classical DAF-1/DAF-4 complex. This example suggests that the TβRII-independent signaling mode might have occurred early in evolution providing different options to fine-tune the TβRI signaling capacity. Furthermore, the recently reported activation of TβRI by exposing glomerular mesangial cells to stretch in the absence of any TGFβ provides further support for the existence of alternative activation modes of TβRI[43].

Taken together these examples support the notion that TβRI has an intrinsic and possibly evolutionary conserved capacity to be constitutively active and that this activity can be amplified/assisted by TβRI mutations or GPR50 by stabilizing the active TβRI conformation.

Aberrant expression and activity of G proteins and GPCRs are frequently associated with tumorigenesis[44]. The ability of GPCRs to promote normal and aberrant cell proliferation often relies on the persistent activation of PI3K/Akt/mTOR, Ras and Rho GTPases, and MAPK cascades[45]. In the case of GPR50, the signaling pathway identified is different since GPR50 promotes the persistent activation of the canonical and non-canonical TGFβ pathways through TβRI transactivation. Our data suggest that GPR50 is a new co-receptor of TβRI substituting TβRII and promoting TGFβ-independent activation of these pathways. In line with the TGFβ-like behavior of GPR50, ectopic expression of

**Fig. 3** GPR50 competes with FKBP12 for the binding to TβRI. **a** (Left) HEK293T cells were transfected with Flag-TβRI alone or cotransfected with HA-TβRII (with TGFβ; 2 ng/mL; 1 h), myc-FKBP12 and either GPR50Δ4 or GPR50wt. Lysates were precipitated for FKBP12 using anti-myc antibody and blotted with an anti-Flag to reveal complex formation. Expression of myc-FKBP12, Flag-TβRI, HA-TβRII and GPR50 was determined in total lysates. (Right) Densitometric analysis of 3 independent experiments (Mean ± s.e.m., $n = 3$ independent experiments, $**p < 0.01$, one-way ANOVA with Dunnett's post hoc test). **b, c** Competition between FKBP12 and GPR50 for TβRI binding was checked by precipitating FKBP12 from total brain and lung lysates of wt and GPR50ko mice and revealed with anti-TβRI. Total lysate was addressed for expression of FKBP12, TβRI, and GPR50 with corresponding antibodies. **d** To address the competition of FKBP12 and GPR50 for TβRI binding, BRET measurements were performed with HEK293T cells expressing fixed amounts of TβRI-Rluc8 and GPR50Δ4-YFP or GPR50wt-YFP or TβRII-YFP and stimulated with TGFβ (0.6 nM, 30 min, 37 °C). Immunoblot on the top shows FBP12 expression when transfected either with empty (Mock) or FKBP12 in different conditions. **e** HEK293T cells expressing the indicated proteins were starved and treated for 1 h with 2 ng/ml of TGFβ or 100 ng/ml of FK506. Total lysates were immunoblotted for Smad3 phosphorylation and total expression of myc-Smad3, GPR50 and myc-FKBP12 with suitable antibodies. **f** Alignment of FKBP12 and GPR50 sequences revealed similarities between the C-terminal "ATGHP" motif in FKBP12 and four repetitive motifs in GPR50 (upper top panel). One motif of GPR50 is located close to the Δ4 deletion of GPR50Δ4 (lower bottom panel). Structural data with permission adapted from Huse et al.[29] highlight the implication of the HP loop (red) in binding to TβRI (lower panel). **g** HEK293T cells were transfected with indicated plasmids and as in **a**. Lysates were precipitated for FKBP12 using an anti-myc antibody and blotted with an anti-TβRI to reveal complex formation. Expression of myc-FKBP12, HA-TβRI, and GPR50 was determined in total lysates. Representative results are shown for **b, c, e,** and **g**. Similar results were obtained in at least two additional experiments. See also Supplementary Fig. 3

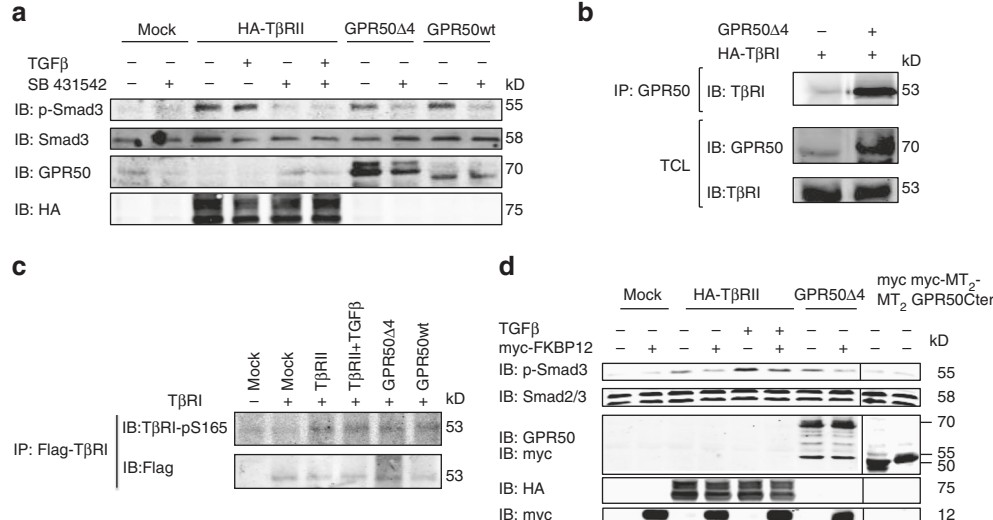

**Fig. 4** GPR50 phosphorylate and activate TβRI independently of TβRII. **a** SNU638 cells were transfected with the indicated plasmids, stimulated for 1 h with TGFβ (2 ng/mL) and pretreated or not overnight with SB431542 at 10 µM. p-Smad3 and total Smad3 levels and expression of transfected plasmids were determined by western blot in cell lysates. **b** Co-immunoprecipitation of GRP50 and TβRI in lysates of SNU638 cells expressing HA-TβRI alone or together with GPR50Δ4. Total lysates were used as expression control (**c**) SNU638 cells were transfected with the indicated plasmids, stimulated TGFβ (2 ng/mL, 1 h). Total TβRI was immunoprecipitated with anti-Flag antibody and immunoblotted for anti-TβRIp-S165. Below, the same blot was immunoblotted with anti-Flag to show the amount of TβRI precipitation in cell lysates. **d** Phospho-Smad3 and Smad2/3 levels were determined in lysates of SNU638 cells stimulated or not with TGFβ (2 ng/mL, 1 h) and expressing the indicated proteins (as verified by western blot). The myc-MT₂ melatonin receptor and the myc-MT₂-GPR50Cter served as negative controls. Representative results are shown for all the panels. Similar results were obtained in at least two additional experiments. See also Supplementary Fig. 4

GPR50 in MDA-MB-231 inhibits cellular proliferation in a TβRI-dependent manner, and tumor development in xenograft model. Inversely, silencing of endogenous GPR50 in NCI-H520 cells promotes cell proliferation and GPR50ko mice show decreased survival after tumor onset and increased tumor growth in the MMTV/Neu spontaneous mammary cancer model. Available data in human cancer data bases indicated that GPR50 under-expression and not overexpression is associated with cancer development and poor survival prognosis indicating that GPR50 predominantly acts as a tumor suppressor. Although most GPCRs seem to have a pro-tumorigenic capacity, some have been reported to exhibit anti-tumorigenic effects similar to GPR50 as shown for the melanocortin 1 receptor in melanoma[46] and the cannabinoid receptors in different cancers[47].

Formation of the TβRI/GPR50 complex adds a further dimension of the regulation of TβRI signaling, which is likely to happen in a cell context-dependent manner. Whereas expression of TβRI is widespread, the expression pattern of GPR50 is more restricted. Expression of GPR50 has been mainly studied in the brain where it has been identified in the pituitary, the dorsomedial hypothalamus, tanycytes, the median eminence and the CA4 region of the dentate nucleus of the hippocampus and the cortex[26,48–50]. Expression of GPR50 in peripheral tissues is less well documented. The GPR50 mRNA has been observed in eye, testis, kidney, adrenal, intestine, lung, heart, ovary, and skin[25].

Significant variation of GPR50 expression has been observed depending on the photoperiod[51], the energy content of the diet and the nutritional status (fed/fasted) of mice[52]. We show here that GPR50 expression is also dependent on TGFβ suggesting a feed-forward regulatory mechanism that might rely on an initial TGFβ-dependent response that, at the long term, is amplified/replaced by a constitutively TβRI response in the TβRI/GPR50 complex. Proteolytic cleavage of the C-tail of GPR50, as reported recently, might be another way to regulate constitutive activation of TβRI[20,38] as the GPR50TM construct, without the C-tail, had no effect on TβRI function. The endogenous ligand for GPR50,

which remains to be identified, might also regulate its activity and that of the TβRI/GPR50 complex.

Differential expression in terms of developmental stages, cell types and tissues are likely to exist between TβRI and GPR50 are likely to exist and maybe at the origin of the embryonic lethality of TβRIko mice as compared to GPR50ko mice. Indeed, TβRIko mice die at embryonic stages and cannot survive after E10.5 because of severe vascular defects[53], whereas GPR50 expression has not been observed (in the vascular system) or starts later (at E13 in several brain region)[54].

In conclusion, we describe here a new molecular complex composed of an orphan GPCR and TβRI that renders TβRI constitutively active towards canonical (Smad2/3) and non-canonical (p38, AKT, ERK) pathways by most likely stabilizing an activated state of TβRI in the absence of TβRII, by phosphorylating TβRI and dissociating the negative regulator FKBP12 from TβRI. GPR50 exhibits TGFβ-like properties, a new tumor suppressor limiting spontaneous mammary tumor development in the MMTV/Neu mouse model and GPR50 underexpression is associated with cancer development and poor survival prognosis.

## Methods

**Cell culture**. HEK293T (Sigma Aldrich-12022001), HeLa (ATCC CCL-2), 4T1 (ATCC CRL-2539), and MDA-MB-231 (ATCC HTB-26) cells were cultured in Dulbecco's modified Eagle medium (GIBCO) containing 10% fetal bovine serum (FCS) (GIBCO) and 1% penicillin/streptomycin (GIBCO). Selective medium containing Genticin (G418) for the maintenance of MDA-MB-231 (250 µg/mL) and 4T1 cell (600 µg/mL) clones (Sigma Aldrich) was used (Sigma Aldrich) was used. SNU638 (Korean cell line bank KCLB No-00638; human gastric carcinoma) and NCI-H520 (ATCC-HTB-182; human lung squamous cell carcinoma) cells were cultured in RPMI-1640 medium (GIBCO) with 10% FCS and 1% Pen/Strep antibiotics. HCMEC/D3 cells were cultured as previously described[55]. Frozen rat tanycytes isolated from rat median eminences were cultured as previously described[56]. Cell lines were checked regularly for any mycoplasma contamination.

**Reagents and antibodies**. *Antibodies:* Phospho-Smad2 (#3108, Cell Signaling Tech. 1/1000), phospho-Smad3(#9520, Cell Signaling Tech. 1/1000), phospho-Stat3 (#9131, Cell Signaling Tech. 1/1000), phospho-Jak2 (#3771, Cell Signaling Tech. 1/

1000), Snail (#3879, Cell Signaling Tech. 1/1000), EEA1 (#C45B10, Cell Signaling Tech. 1/100) and total Stat3(#4904, Cell Signaling Tech. 1/1000) were purchased from Cell Signaling Technology; SMAD2/3 (610843, BD Biosciences, 1/1000), anti-myc (sc789, 1/1000 and 9E10, sc40, 1/1000), TβRI (sc398, 1/500) Lamin B (sc6217, 1/1000) TβRII (sc17792, 1/500), FKBP12 (sc6174, 1/500), GAPDH (sc32233, 1/1000) and GPR50 (sc50590, 1/500) came from Santa Cruz Biotechnology. Monoclonal and polyclonal Flag-Antibodies were used from Sigma-Aldrich (M2-F3165, 1/1000, and F7425, 1/1000). HA (11867423001, 1/1000) and GFP (11814460001, 1/2000) antibodies were used from Roche. Anti-Tubulin was purchased from AbD Serotec (MCA77G, 1/2000). GPR50 antibody7 was produced by Kernov Antibody Services (1/1000)[49]. Polyclonal Anti GPR50 (H00009248-B01P, 1/500) was purchased from Novus Biologicals. Anti TGFβRI-pS165 (ab112095, 1/500) was purchased from Abcam. All antibodies were employed according to the recommended dilutions for either immunoprecipitation or western blotting.

*Reagents:* FK506 and SB431542 were purchased from Sigma Aldrich and recombinant TGFβ-1 was purchased from Peprotech.

**Cell transfection and generation of stably overexpressing cells.** Transient transfection was performed by using Lipofectamine® LTX reagent (Life Technologies) in HEK293T cells, Xtremegene® 9 (Roche) for reporter gene assay in HeLa cells and jetPRIME (Polyplus) reagent for MDA-MB-231 and SNU638 cells, each employed according to the manufacturer's instruction. Cells were incubated for 48 h before experimental use. Stably GPR50 overexpressing cells were generated by jetPRIME Transfection of G418 resistant GPR50 plasmid. Selective pressure was established by using conditioned DMEM medium with 1 mg/mL G418 (Sigma Aldrich). Stably transfected 4T1 cell lines were established by selection with neomycin (G418) antibiotic. Monoclonal cell lines were obtained through a dilution limit process and positive clones expressing empty plasmid, GPR50Δ4 and GPR50wt were identified with western blot.

**Animals and tissue extraction.** All experiments were carried out in accordance with the European Communities Council Directive of 24 November 1986 (86/609/

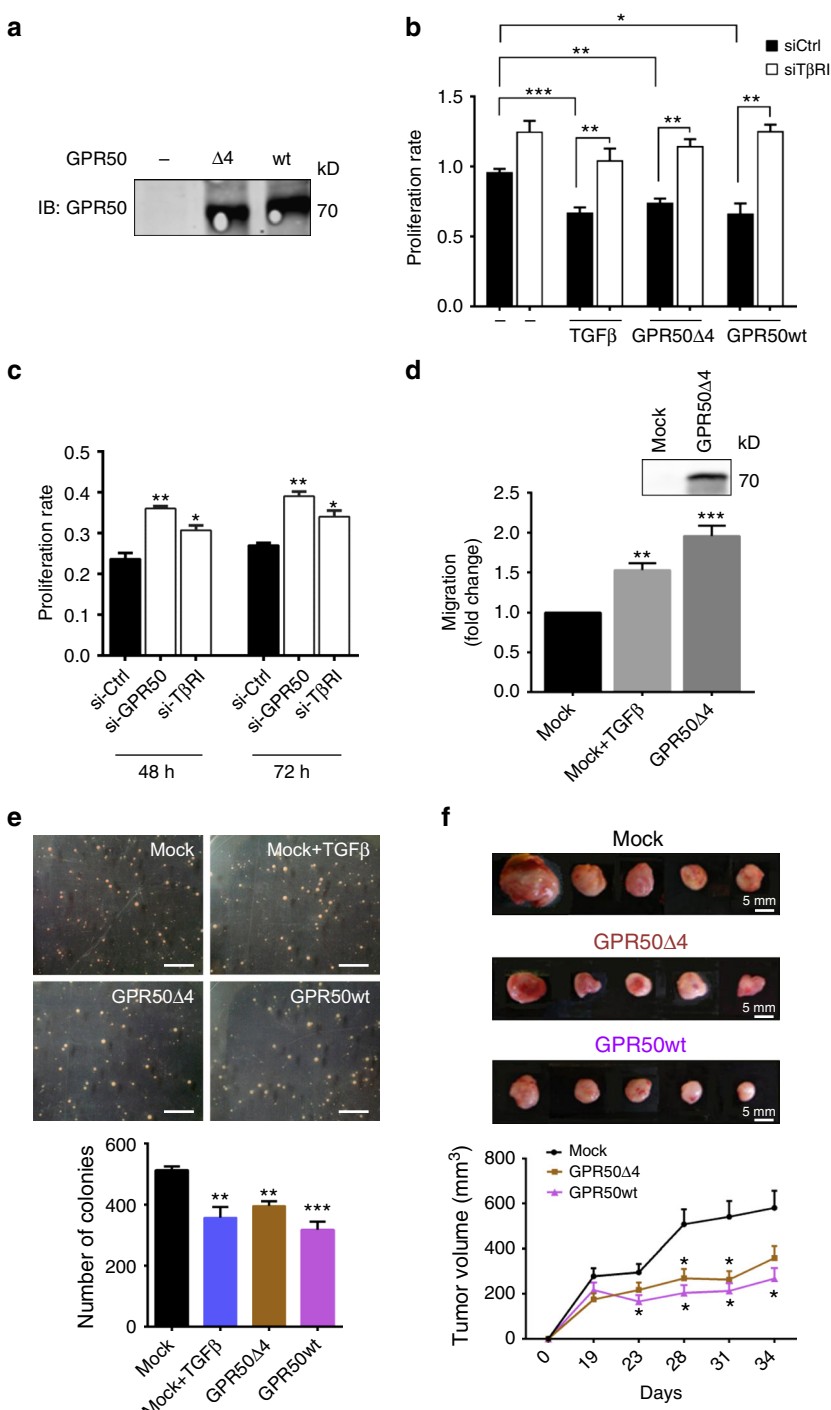

EEC) and the experimental protocols were approved by the local institutional research animal committee. For coIP experiments tissues were taken out from both, C57BL/6J wildtype and GPR50ko mice, homogenized and tissue was solubilized in 0.5% CHAPS detergent in TEM buffer (25 mM Tris pH7.4; 2 mM EDTA; 10 mM MgCl$_2$) on wheel for 6 h at 4 °C. The solubilized samples were centrifuged at 14,000 rpm for 45 min at 4 °C and supernatants were collected. Further, the lysates were processed either for immunoprecipitation or immunoblotting. For immunoprecipitation lysates were incubated with either anti-GPR50 or anti-TβRI antibody overnight and followed by coIP procedures as described in section of immunoprecipitation for cells.

**Plasmid mutagenesis**. Primers for point mutations were designed with the help of the Agilent QuikChange Primer Design program. Mutagenesis was performed by PCR with the Phusion High Fidelity Polymerase (Finnzymes, Thermo-Fisher Scientific).

**Co-immunoprecipitation and western blotting**. For preparation of cellular lysates, cells were harvested after transfection and stimulation according to protocol, in TNMG buffer with 0.5% NP-40[57] for 15 min, centrifuged at maximum speed for 30 min. and supernatants were kept. Lysate containing 500 μg to 1 mg protein were subjected to immunoprecipitation by incubating for 3 h with 2 to 5 μg of antibody. Protein G beads (Sigma), were added for additional 2 h, prior to three washing steps in a Tris-EDTA-magnesium buffer with 0.05% NP-40 buffer. Samples were diluted in 2× Laemmli with 4% SDS and heated for 5 min at 95 °C preceding SDS-PAGE. Cell Lysates for protein analysis were obtained after lysis with TNMG buffer, protein estimation was performed with BCA Assay (Thermo-Fisher Scientific), 20 to 50 μg of sample were prepared and 4× SDS-Laemmli was added. Samples were heated at 95 °C and separated on a 12% acrylamide SDS gel-electrophoresis. Proteins were blotted on a PVDF membrane (Dutscher), blocked and incubated overnight with antibodies in 3% skimmed milk or BSA-solution. Incubation with fluorescence coupled secondary antibodies enables readout on an Odyssee reader. Densitometric analysis was performed using ImageJ software. The uncropped blots showing cropped area of the most important blots are provided as supplementary information (Supplementary Fig. 7-13). GPR50 was detected at 50kD in tissues by Anti GPR50 (H00009248-B01P; Novus Biologicals) than usual size at 70 kD by GPR50 antibody 7.

**Cross linking experiment**. HEK293T cells were lysed in TNMG buffer, except for using 20 mM instead of 20 mM Tris-HCl. Cell extracts were incubated with 0.5 mM of DSS (disuccinimidyl suberate; Thermo-Fisher Scientific) for 30 min. at room temperature. The reaction was then quenched with 1 M Tris (pH 7.5), at a final concentration of 50 mM and incubated for an additional 15 min at room temperature. Samples were then subjected to immunoprecipitation using the anti-Flag or anti-HA antibodies, followed by immunoblotting with anti-Flag or anti-HA antibodies.

**Nuclear extracts**. HEK293T cells were seeded in 100 mm culture plates and transfected with mock, Smad3 alone or cotransfected with GPR50wt and GPR50ΔTTGH variant. The cells were starved for 16 h in DMEM media without FBS and stimulated with 2 ng/mL of TGF-β for 2 h. The culture plates were rinsed twice with ice-cold phosphate-buffered saline (PBS). An aliquot of 500 μL of hypotonic buffer containing 1% NP-40 was added to each culture plate, and allowed to swell on ice for 15 min. The cells were scraped and taken into fresh Eppendorf tube. The lysate was vortexed for 10 s, and the nuclei were pelleted (14000 rpm for 1 min). Supernatant was collected and referred to as "cytoplasmic fraction". The nuclear pellets were resuspended in 100 to 200 μL of hypertonic buffer and rotated for 30 min at 4 °C. This extract was then centrifuged (14000 rpm for 20 min), and the supernatant collected referred to as "nuclear fraction". The

amount of protein was estimated following BCA estimation kit (Thermo-Fisher Scientific). The buffer compositions were as follows (i) Hypotonic buffer contained 10 mM HEPES (pH 7.9), 10 mM KCl, 1.5 mM MgCl$_2$, 1 mM EDTA, 25 mM β-glycerophosphate, 1 mM Na$_3$VO$_4$, 1 mM dithiothreitol (DTT) (ii) Hypertonic buffer contained 20 mM HEPES (pH 7.9), 420 mM NaCl, 1.5 mM MgCl$_2$, 25% glycerol, 1 mM EDTA, 25 mM β-glycerophosphate, 1 mM Na$_3$VO$_4$, and 1 mM DTT. To both buffers protease and phosphatase inhibitors were added just before use.

**Reporter gene assay**. HeLa cells were seeded in 12-well-plates transfected with the SMAD2-dependent Activin-response-element (ARE) Firefly Luciferase or the SMAD3-dependent CAGA Firefly luciferase reporter gene, increasing amounts of GPR50 ΔTTGH or WT and *Renilla* Luciferase as internal standard. Cells were left for 24 h starved and stimulated with TGFβ (0.5 ng/mL, 8 h). Lysis and measurement were performed with the Dual Luciferase Kit (Promega) according to manufacturer's advices. Experiments were performed in triplicates, figures show representative experiments.

**MTT proliferation assay**. In a 96-well flat-bottomed plate 1000 cells/100 μL of DMEM were seeded. Following serum starvation cells were stimulated with TGFβ (2 ng/mL) and MTT, a tetrazolium dye (Sigma-Aldrich) was added to each well after 1, 3, and 5 days of TGFβ stimulation. After adding MTT, plates were incubated at 37 °C for 4 h, finally 100 μL of DMSO was added to each well to solubilize the purple colored formazan crystals and absorbance was recorded at 560 nm on a microplate reader.

**Immunofluorescence**. Cells were fixed in 4% paraformaldehyde for 15 min permeabilized with Triton X100 (0.3%) for 5 min and then blocked with a 5% horse serum/PBS solution for 1 h. Cells were incubated overnight with primary antibody, fluorescent secondary antibodies (Life Technologies) were added and slides were visualized by confocal microscopy.

**Proximity ligation assay**. Cells were fixed in 4% paraformaldehyde for 15 min. permeabilized with Triton X100 (0.3%) for 5 min and then blocked with a 5% Horse Serum/PBS solution for 1 h. Incubation with primary antibodies (anti-GPR50 goat and anti-T βRI rabbit) was done at 4 °C overnight. PLA was performed by the use of a Duolink® In Situ Red Starter Kit Goat/Rabbit (Sigma-Aldrich) according to the manufacturer's protocol. Results were visualized on a confocal microscope. For negative controls, the whole process was done with only one primary antibody.

**TβRI kinase assay**. For the kinase assay, protein lysates of HEK293T cells transfected with Mock- or GPR50 plasmid were incubated with a GPR50 antibody. Precipitates were separated with Protein G Sepharose (Sigma) Aldrich, washed three times in lysis buffer TNMG with 0.05% NP40 and one time in "reaction buffer". Precipitates were incubated with a TβRI Kinase (TGFβR1 Kinase Enzyme System, Promega) and kinase activity was measured by incubating with the delivered kinase substrate (Smad3 Peptide) and the use of the ADP-Glo™ Kinase Assay kit (Promega) according to the manufacturer's instructions.

**Tandem affinity purification**. Tandem affinity purification-mass spectrometry-(TAP-MS) captured proteins were washed and digested on calmodulin beads by trypsin for 3 h at 37 °C in 50 mM ammonium bicarbonate after reduction in 20 mM DTT at 60 °C for 30 min and alkylation in 50 mM iodoacetamide in the dark at RT for 30 min. The supernatant was collected for nLC MS/MS analysis and treated as described[58] but with following settings for protein identification:

---

**Fig. 5** GPR50 inhibits cell proliferation and tumor growth in MDA-MB-231 cells. **a** Expression of GPR50Δ4 and GPR50wt in lysates of MDA-MB-231 cell pools revealed by western blot. **b** An equal number of MDA-MB-231 cells expressing Mock, GPR50Δ4 or GPR50wt were seeded into 96-well plates, starved and stimulated with TGFβ (2 ng/mL) and transfected with either siRNA against control (si-Ctrl) or TβRI (si-TβRI). The proliferation rate over total amount of cells was measured with the MTT assay 5 days after starvation (Mean ± s.e.m., $n = 5$ independent experiments, *$p < 0.05$, **$p < 0.01$,***$p < 0.001$ one-way ANOVA with Tukey multiple comparison post hoc test). **c** Proliferation rate of NCI-H520 cells at 48 h and 72 h after transfecting control si-RNA (si-Ctrl), GPR50 si-RNA (si-GPR50) or TβRI si-RNA (si-TβRI). The graph is representative of five independent experiments. (Mean ± s.e.m., $n = 5$ independent experiments, *$p < 0.05$, **$p < 0.01$, one-way ANOVA with Dunnett's post hoc test). **d** A wound healing assay and live-cell imaging was used to assess the cell migration properties of 4T1 cells stably expressing either empty vector (Mock) or GPR50Δ4 plasmid in presence or absence of TGF-β during 24 h. Top inset blot shows expression of GPR50Δ4 in 4T1 stable cells (Mean ± s.e.m., $n = 3$ independent experiments, **$p < 0.01$, ***$p < 0.001$ one-way ANOVA with Dunnett's post hoc test). **e** Anchorage-independent growth assay of MDA-MB-231 cell pools expressing Mock, GPR50Δ4 or GPR50wt was monitored for 3 weeks, stimulation with TGFβ was done once a week (2 ng/mL). Images in upper panel show an example of colony number and distribution. The lower panel histogram shows the mean value ± SEM of the colony number of four dishes for each condition in one representative experiment (Mean ± s.e.m., $n = 3$ independent experiments, **$p < 0.01$,***$p < 0.001$, one-way ANOVA with Dunnett's post hoc test). **f** Xenograft experiment after injection of MDA-MB-231 cell pools into the flanks of nude mice. Images in upper panel show five representative tumors (5/10 for MOCK and GPR50wt and 5/8 for the GPR50Δ4). The graph in the lower panel shows tumor growth during 34 days (Mean ± s.e.m., $n = 5$, *$p < 0.05$, two-way ANOVA with unpaired $t$-test). See also Supplementary Fig. 5

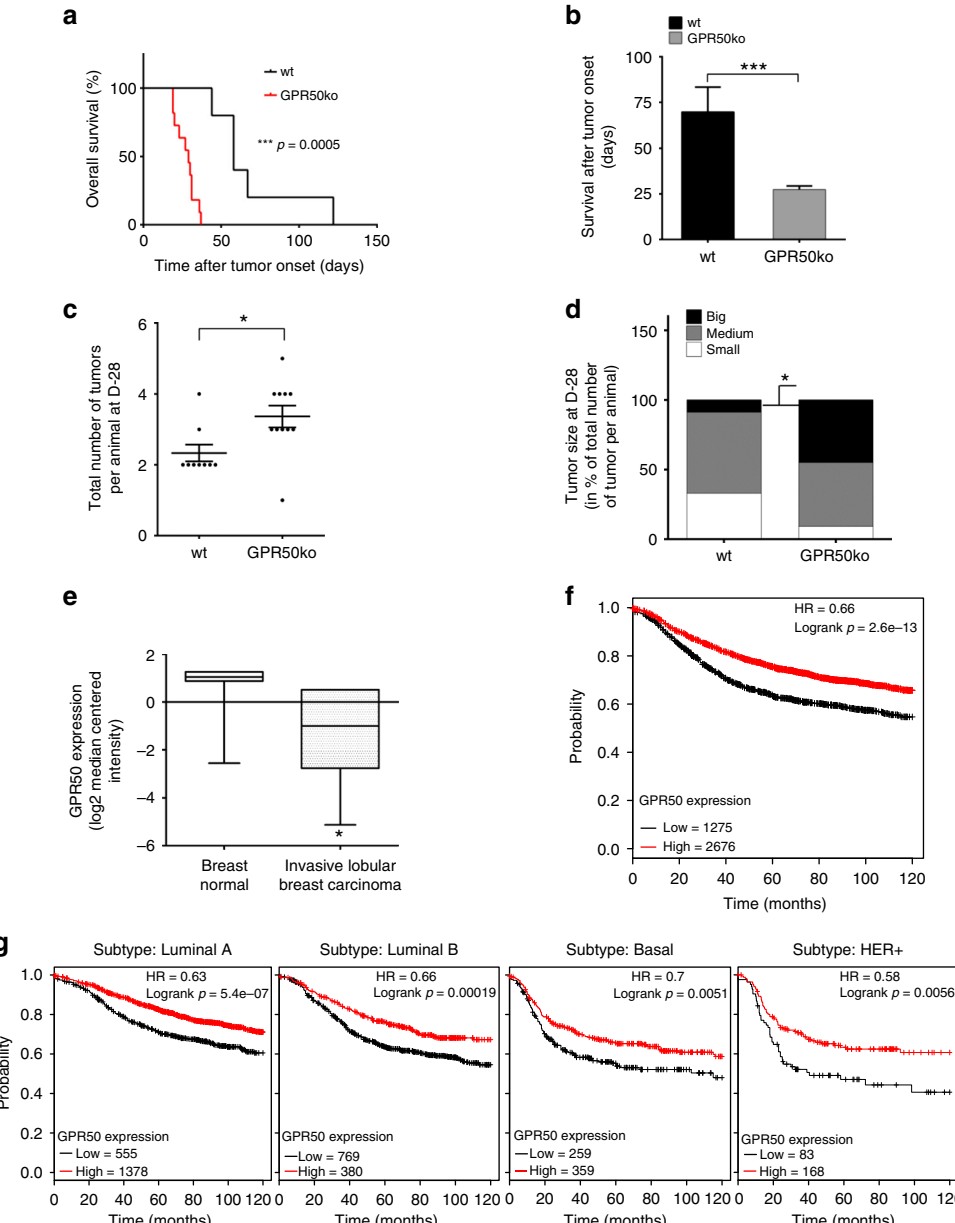

**Fig. 6** Knockout of GPR50 in the MMTV/Neu mouse model. **a, b** Survival curves of MMTV-Neu-wt and MMTV-Neu-GPR50ko mice where (**a**) represents overall survival onset (Mean ± s.e.m., WT $n = 5$, ko $n = 11$, Kaplan–Meier survival curve analysis with log-rank Mantel–Cox test, ***$p < 0.001$) and (**b**) average survival after tumor (Mean ± s.e.m., WT $n = 5$, ko $n = 11$, unpaired two-tailed $t$-test ***$p < 0,001$). **c, d** Total number of tumors (**c**) and size distribution of tumors (**d**) in MMTV-Neu-wt and MMTV-Neu-GPR50ko mice at day 28 after tumor onset (average survival day of MMTV-Neu-GPR50ko mice) (**c** mean ± s.e.m., WT $n = 9$, ko $n = 11$, unpaired two-tailed $t$-test *$p < 0.05$ and **d** mean ± s.e.m., WT $n = 9$, ko $n = 11$, two-way ANOVA with Bonferroni's test, *$p < 0.05$). **e** Box plot showing GPR50 mRNA expression in normal and cancerous breast tissue. Data analysis (unpaired $t$-test) was performed with the oncomine 3.0 database (oncomine.org; Redvanyi breastdatabase-U52219 reporter)[58, 61]. **f, g** Kaplan–Meier survival curves for high versus low GPR50 mRNA expression in breast cancer were generated using an integrated database and online tool (http://kmplot.com/breast/)[62]. Probability of relapse-free survival in high (red) or low (black) GPR50 expressing breast cancer is compared across all tumors (**f**) and according to molecular subtypes of breast cancer (**g**). See also Supplementary Fig. 6 and Supplementary Tables 1-6

Extracted MS/MS peak-lists were submitted to an in-house Mascot (Matrix Science), version 2.2, search engine. The Swiss-prot database (28 April 2008, 366,226 sequences; 132,054,191 residues) was used restricted to the Homo sapiens subset of sequences (19,372 sequences). Parent and fragment mass tolerances were, respectively, set to 100 p.p.m. and 0.3 Da, partial modification (oxidization) of methionines was allowed. Missed trypsin cleavage sites were limited to 1. A filter was applied to the search to reduce false positives and matching redundancies of the same peptide in several hits. All peptide matches above 1% risks of random matching were eliminated ($p < 0.01$ filtering was applied). The individual minimum peptide score was set to 30. False positives rates were evaluated using Mascot. Identification was considered successful if at least two distinct peptides were identified. Under these stringent parameters, the minimum protein score was 29[59].

**BRET analysis**. HEK293T cells were transiently transfected in six-well plates with 100 ng or 100 to 2000 ng respectively of the corresponding Luciferase-coupled and YFP-coupled plasmids for competition experiments with constant amount of Luciferase-coupled and YFP-coupled plasmids and increasing doses of myc-FKBP12. Cells were grown overnight and transferred into 96-well-Optiplates (PerkinElmer Life Sciences), pre-coated with 10 µg/mL poly-L-lysine (Sigma), where they were grown for additional 24 h, washed with PBS. Coelenterazine (Interchim France) for Luciferase stimulation was added and cells were subjected to measurement of emission at Luc and YFP wavelength on a Berthold Mithras™ as previously described[60]

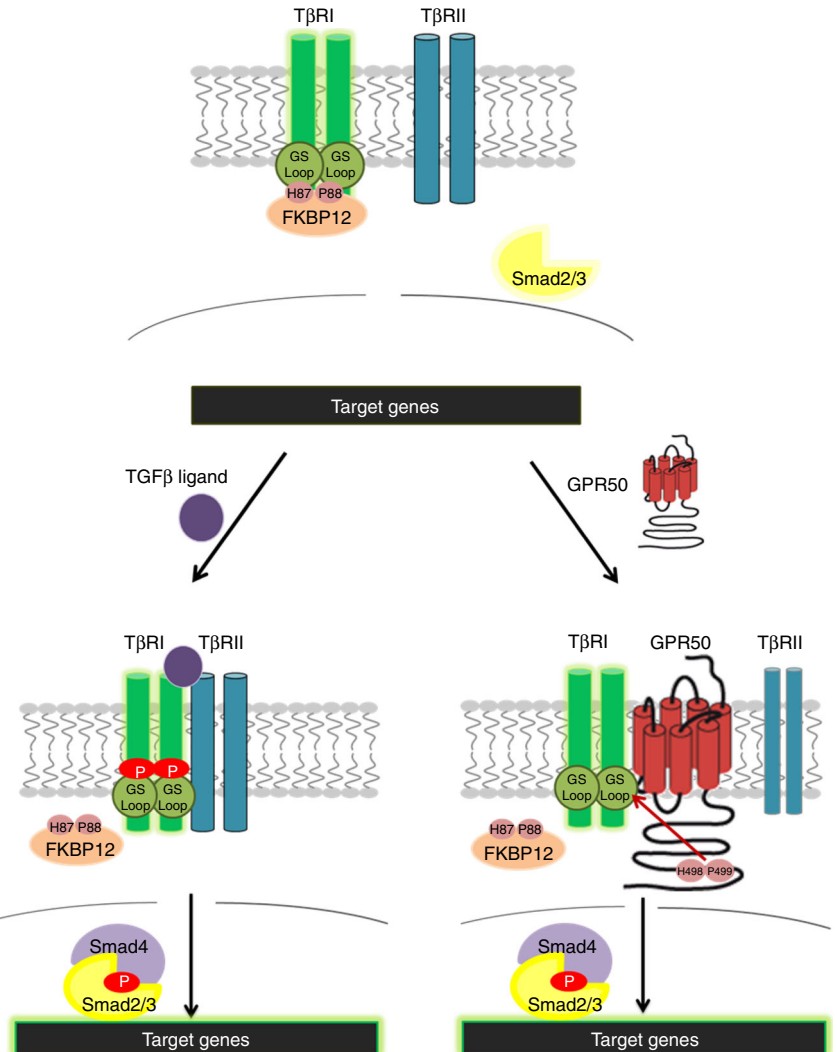

**Fig. 7** Proposed Model of GPR50 action on TβRI-mediated signaling In the basal state (upper part), TβRI and TβRII, each form homodimers, that are apart from each other. TβRI is stabilized in its inhibitory confirmation by FKBP12. The R-Smads are non-phosphorylated in the cytosol and no transcription of target genes occurs. In the classical activation mode (lower left side), TGFβ binds to TβRII, which enables recruitment of TβRI into the complex. TβRII phosphorylates TβRI in the GS domain, FKBP12 dissociates from the complex and R-Smad2/3 becomes phosphorylated by TβRI after recruitment into the complex. Phosphorylated R-Smad2/3 dissociates and forms a complex with Smad4, which translocates into the nucleus, and regulates target gene expression. In the case that TβRI forms a complex with GPR50 (and not with TβRII) (lower right side), GPR50 induces the dissociation of FKBP12 from TβRI due to a similar motif of amino acids H87 and P88 in its C-tail on positions 498 and 499. The GPR50/TβRI complex then constitutively activates the classical and non-canonical Smad signaling pathways

**Radioligand-binding assay.** MDA-MB-231 cells were plated in 12-well plates in incubated with $^{125}$I-TGFβ (100,000 cpm/mL; NEX267) in DMEM, 20 mM Hepes, pH 7.4, 0.4% BSA for 4 h at 4 °C to determine the number of surface exposed receptors. Non-specific binding was determined in the presence of a 100-fold excess of unlabeled TGFβ. Cells were washed twice with ice-cold PBS and extracted in 1 mL 1 N NaOH and $^{125}$I-TGFβ quantified in a scintillation counter.

**Generation of GPR50ko/MMTV/Neu mice and sample collection.** All procedures involving animals were performed with the approval of the CEEA (Animal Experimentation Ethical Committee) according to the EU official regulations. Generation of the congenic strain GPR50ko/MMTV/Neu was accomplished by mating FVB/N-Tg(MMTVneu)202Mul/J mice from Jackson Laboratory with the GPR50ko C57Bl/6 mouse strain and backcrossed into the FVB background. Females (wt, $n = 21$ and ko, $n = 19$; FVB background) were palpated weekly for mammary gland nodules. The animals were observed from 4 months till 2 years. After 2 years, all surviving animals were sacrificed. The time of first tumor detection was defined as "0" for the calculation of the "Time after tumor onset for each animal. As soon as the tumors appeared they were routinely measured with external caliper and volume was calculated as $(4\pi/3) \times (width/2)2 \times (length/2)$. Depending on the group, animals were killed at the following time points: (1) when the end point was reached (20 mm of apparent tumor diameter), (2) 28 days after

primary tumor detection (average survival of GPR50ko mice), and breast tumors were collected. Size of the tumor presented as; small-tumor = non detectable by palpation $d < 2$ mm; medium-tumor = detectable by palpation but non visible 2 mm $\leq d < 5$ mm; big-tumor = visible 5 mm $\leq d$.

**Generation of GPR50ko mice.** The targeting constructs for GPR50ko mice were generated from PCR products amplified from the DNA of 129/SV ES cells with Pfx polymerase (Life technologies). For the targeting vector constructs, LoxP sites were introduced upstream and downstream of exon 2 of the GPR50 gene, and a hygromycin cassette flanked by FRT sites was introduced as a selection marker downstream of exon 2 of GPR50. The targeting constructs were introduced by electroporation into embryonic stem cells from the 129/SV mouse strain and selected on plates containing hygromycin. Appropriate clones were identified by PCR and confirmed by Southern blot analysis with 5′, 3′ and an internal probe. A karyotype and the sequencing of the homologous recombinant clones have been done. Stem cells carrying the constructs were injected onto blastocysts from C57BL/6 N mice, to obtain chimeric mice. After germline transmission, the generated GPR50$^{f/f}$ mice were initially crossed with a germline Flp-deleter murine line to eliminate the hygromycin cassette and then subsequently crossed with the germline EIIa-cre murine line in order to generate the GPR50ko mice model.

**Soft agar assay or anchorage independence assay**. Dishes of 35 mm dishes were coated with a layer of 0.5% Agar containing a DMEM/7% FBS solution. A total of $1 \times 10^5$ MDA-MB-231 cells were mixed with 0.3% agar containing DMEM media with 7% FCS and distributed upon the first layer. Colony formation was measured about 20 days after seeding.

**Xenograft experiments**. A total of $1 \times 10^6$ MDA-MB-231 cells were diluted 1:1 in a Matrigel™ (BD Biosciences) and subcutaneously injected into the right and left flank of nude mice. Tumor growth was monitored every 3 days and measured using a caliper. Tumor volume was calculated with the $(width)^2 \times length \times \pi/6$ formula.

**Cell migration assay**. 4T1 stable clones were seeded on Culture Inserts (Ibidi). Ibidi inserts were removed and closure of the resulting wound was monitored over the next 24 h. Images of the wound were captured using the Olympus IX83 microscope system equipped with Phase contrast objective. Three spots were chosen in each plate to compute the healing rate. Images were captured in Cell sens dimension v1.16. Gap area measurement was calculated with the FIJI software and a semi-automated macro.

**Analysis of public breast cancer microarray data sets**. The probability of relapse-free survival according to GPR50 status analyses was generated using the Kaplan–Meier Plotter, an integrated database and online tool (http://kmplot.com/analysis)[60]. The patients were divided into high versus low GPR50 expression groups using the auto-select best cutoff. Survival curves were generated across all tumors or according to molecular subtypes of breast cancer. The hazard ratio with 95% confidence intervals and log-rank $p$ value are calculated for low versus high GPR50 expression. Further, GPR50 expression analysis was carried out in normal and cancerous tissue by using oncomine online database (oncomine.org).

**Statistical analysis**. Data were analyzed using GraphPad Prism 6 software (GraphPad Software) and are presented as mean with s.e.m. Statistic tests were performed and $p$ value thresholds were obtained using GraphPad 6. Multiple groups were tested using analysis of variance (ANOVA) and comparisons between two groups were performed using two-tailed unpaired Student's $t$-test. To analyze MMTV/Neu mice experiment, two-way ANOVA followed by bonferroni's post hoc test and Kaplan–Meier survival curve analysis was performed by using Graph-Pad prism. Statistically significant differences are indicated as follows: $^*p < 0.05$; $^{**}p < 0.01$; $^{***}p < 0.005$; $^{****}p < 0.001$. Representative figures are shown in Figs. 1b–h,j, 2a-d, e, f, h, i, 3a, b, c, e and g, 4a–d and Supplementary Figs 1d-h, 2a-e, and 2g-j, 3a, c, e and 3f, 4a, b, and 4d, 5a, c and 5d. Each experiment was repeated independently at least twice.

**Data availability**. The public breast cancer survival and expression data referenced during the study are available in a public repository from the http://kmplot.com/analysis, prognoscan.org, and oncomine.org websites. The authors declare that all the other data supporting the findings of this study are available within the article and its supplementary information files and from the corresponding author upon reasonable request.

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

## Acknowledgements

We are grateful to Dr Mark Scott (Institut Cochin, France) for expert advice and Cédric Broussard from the Plateform Protéomique 3P5 for protein identification by mass spectrometry. We thank Romain Morichon, UMS LUMIC, CRSA for videomicroscopy, real-time cell migration analysis and Anisia Silva (Institut Cochin, Paris) for technical help. This work was supported by grants from the Fondation Recherche Médicale (Equipe FRM 2006 to R.J.), ARC N° N°SFI20121205906, Agence Nationale de la Recherche (ANR-16-CE18-0013 to J.D.) and the "Who am I?" laboratory of excellence No. ANR-11-LABX-0071 funded by the French Government through its "Investments for the Future" program operated by The French National Research Agency under grant No.ANR-11-IDEX-0005-01 (to R.J. and R.A.), Inserm and CNRS. S.W. was supported by a doctoral fellowship from the CODDIM 2009 (Région Ile-de-France) and K.T. by a research fellowship from the Université Paris Descartes.

## Author contributions

Conceptualization: S.W., R.A., Z.B.-C., A.D., P.D., C.P., and R.J.; investigation: S.W., R.A., Z.B.-C., A.-S.J., S.G., J.D., A.D., D.N.L., O.L., A.K., A.S., N.C., T.C., N.K., K.T., R.J. Resources: V.P., P.D., M.D.C., F.G.; writing of the original draft: S.W. and R.J.; writing—review and editing: S.W., R.A., A.-S.J., J.D., A.D., C.P. and R.J.; funding acquisition: P.D., J.D. and R.J; Supervision: R.A., J.D., J.-L.G., N.C., P.D., V.P., O.H., C.P. and R.J.

## Additional information

**Competing interests:** The authors declare no competing interests.

