## [Peer Review File(PDF 328 kb) · Nature Communications]

Reviewers' comments:

Reviewer #1 (Remarks to the Author):

In this work Wojciech et al. have identified a new mechanism that regulates the activity of the TGF-beta receptor I (TbRI), which does not require interaction with the TGF-beta receptor II (TbRII), but the formation of an alternative complex between TbRI and the orphan GPR50, a member of the G-coupled receptor superfamily.

Work shows great novelty and interest, not only due to the relevance of the new mechanism described, but also because authors results suggest that this interaction might be altered in tumors that would loss the response to TGF-beta as a tumor suppressor.

Experiments are well designed and performed and most of the conclusions are supported by the data. However, some points must be taken into consideration before publication of the manuscript:

1. It is not completely clear how the interaction between GPR50 and TbRI favors activation of the TbRI kinase capacity. One of the canonical mechanisms that activate TbRI is through phosphorylation by TbRII. Authors propose that GPR50 activation does not require TbRII. But which is the level of phosphorylation of the TbRI in the absence or presence of GPR50 expression? Is GPR50 provoking TbRI phosphorylation by alternative mechanisms that do not require TbRII? Alternatively, is the increase in Smad2 and Smad3 phosphorylation, observed after GPR50-TbRI interaction, independent of the level of TbRI phosphorylation? Authors suggest potential allosteric modifications in the TbRI as consequence of the GPR50 interaction, but they do not demonstrate how it could affect to TbRI phosphorylation. At least, authors must explore the phospho-TbRI levels by performing experiments in cells lacking TbRII, such as those used in the manuscript, after overexpressing the active form of GPR50.

2. Another alternative possibility that would explain the observed effects should be that interaction between GPR50 and TbRI would induce changes in the TbRI distribution in the cell membrane, moving it to early endosomes that would favor its interaction with SARA or other Smad2/3 scaffolds. Authors mention that they explored the TGF-beta receptor cell surface expression, but results are not shown. Furthermore, the most interesting point should be to explore potential differences in the TbRI intracellular trafficking when GPR50 is overexpressed.

3. Authors mainly focus on the suppressor effects of the TGF-beta, ignoring the numerous situations where this cytokine contributes to promote tumor dissemination, migration and metastasis, particularly by its capacity to induce epithelial-mesenchymal transition. This work would require that authors present some data indicating whether or not the interaction of GPR50 with TbRI may affect to the non canonical pathways induced by TGF-beta in some tumor cells. A cancer cell line that responds to TGF-beta inducing migration/invasion, but not cytostasis, must be chosen, to explore whether this GPR50- TbRI interaction exists and which are the effects on the response to TGF-beta. It was described that phosphorylation of Smad3 is required for Snail upregulation by TGF-beta. It should be expected that GPR50, through activating TbRI, could be favoring TGF-beta pro-tumorigenic actions, too. But it is necessary to demonstrate it. Non-canonical signals involved in TGF-beta pro-tumorigenic effects, such as PI3K/AKT and ERKs, must be considered to be explored in this study too.

Reviewer #2 (Remarks to the Author):

GPR50 is an orphan GPCR that is highly related to the two melatonin receptors, MT1 and MT2, but does not bind melatonin, and the GPR50 endogenous ligand is still unknown. It was initially shown that GPR50 interacts with and limits the activation of MT1 receptors. In a yeast-two-hybrid screen, it was found that GPR50 also acts as a signaling partner and modulator of the transcriptional co-activator TIP60, thereby regulating glucocorticoid receptor signaling. Most of these interactions occur through the large carboxyl terminal tail (C-tail) of GPR50, which functions as a scaffold for interacting partners. Using proteomics approaches, the authors have now found that GPR50 binds to T β RI through its C-tail, and that this results in increased T β RI signaling independently from T β RII. At the molecular level, evidence is presented that GPR50 enhances the basal TGF β -independent function of T β RI, increasing Smad2/3 phosphorylation likely by competing with the inhibitory binding of FKBP12 to T β RI and by stabilizing the active T β RI conformation. Evidence is also presented that GPR50 has antitumor activity in mouse models, and that GPR50 levels are associated with poor survival prognosis in selected human breast cancer studies. Overall, there is some interest in this study, which involves a large repertoire of strategies to document the interaction between T β RI and GPR50 and its functional consequences. That said, although statistically significant many of the effects appear to be subtle, and difficult to compare to the effects achieved by TGF β by acting on the T β RI-T β RII complex. While the interaction between GPR50 and T β RI can occur, this reviewer is not convinced about the biological relevance of this observation.

What is the stoichiometry of GPR50 binding to T β RI with respect to T β RI binding to T β RII?

The initial interaction of GPR50 with T β RI was identified by tandem affinity purification coupled to mass spectrometry using the GPR50 Δ 4 variant in HEK293T cells. What were the statistical methods used to determine the potential significance of the identification of 5 unique peptides corresponding to the T β RI in their analysis? How does it compare to the other peptides identified during this study?

In Figure 1, in which the interaction is first documented, many of the panels lack appropriate controls. For example, in panel e, the control IgG IP appears to have been spliced out from the IP with anti GPR50. It is strange that the authors do not show T β RI binding to T β RII in the T β RI IP, and the bands on the total cell lysates one has to trust that they represent the appropriate proteins without the benefit of controls, for example of knock down or knock out cell lysates.

Same of the association between the GPR50 Δ 4 and GPR50 to T β RI in BRET assays when compared to T β RI association to T β RII.

The analysis of increased expression of GPR50 in response to TGF β in NCI-H520 cells is pedestrian at its best. No explanation or mechanistic information is presented, and the authors make very strong statements of a negative feedforward loop without much data supporting it.

What is the rationale for the selection of NCI-H520 cells for this studies, and how many other lung cancer cells express this receptors?

In Figure 2 a, why they need to over expressed Smad3? In Figure 2 b, one would have expected that the authors would have used at least 2 siRNAs, and to use TGF β stimulation and T β RI knock down as controls. The latter is a concern thought the study, as it is hard to judge whether many of the subtle changes caused by GPR50 expression or knock down/out are meaningful (in addition to statistically significant) with respect to those caused by canonical activation of the pathway. This is clear, for example, in Figure 2 e, in which nuclear pSmad2/3 is barely affected by GPR50 Δ 4 expression, and almost nothing by GPR50. Same for reporter activation (Figure 2 f and g).

In Figure 3 a, it is hard to tell whether GPR50 really competes for FKBP12 binding to T β RI, and in every case this competition is observed after large overexpression. The data should be compared

to the effect of TGF β by acting on T β RI/II. Same for the normalized BRET association and T β RI kinase activation assays, which when normalized to control lysate is quite limited, making it difficult to assess its potential biological relevance.

In Figure 5 c, the increased proliferation rate promoted by knockdown of GPR50 is greater than that caused by T β RI, which when aligned with the comments above, suggest that at least part of GPR50 biological functions are independent from T β RI. The usefulness of the over expression experiments in Figure 5 b is questionable. It is somehow surprising that the authors have used mock transfected cells for this and other tumor studies and biological experiments. Were mock transfected cells subjected to similar selection than GPR50/ GPR50 Δ 4 stably expressing cells (G418 resistant)? This may altered cellular behavior, skewing the cell populations used for the studies.

In Figure 6, it is unclear what is the meaning of "time after tumor onset", as these mice develop tumors quite late in life, when around 200 days old, and there was no difference in tumor free survival between wt and GPR50 KO mice (sup Fig 6 d). Were these control littermates?

With thousands of breast cancer patients studied in TCGA and other efforts by RNAseq, it is unclear why the authors cherry picked some studies using gene array technologies that were conducted prior to this new database. Indeed, Prognoscan.org was last updated in 2012, before TCGA, and datasets are quite old as well.

Few additional comments:

Is GPR50 expressed in immune cells? This may affect tumor development independent from TGF β signaling.

Are G proteins involved or all effects are G protein (or arrestin) independent?

Given the rest of the available information regarding the proposed central role of the association of the GPR50 C-tail with T β RI, the authors may want to explain their interpretation of the data that cytosolic C-tail (GPR50Cter) alone or fused to the 7TM domain of the melatonin receptor MT2 (MT2-Cter GPR50) are not able to replicate the limited spontaneous activation of the Smad pathway.

Mice KO for T β RI exhibit a remarkable phenotype, while GPR50 KO mice are viable and exhibit subtle phenotypes. This seems to represent a disconnect that has not been addressed and/or discussed.

Response to Reviewers' comments:

Reviewer #1 (Remarks to the Author):

In this work Wojciech et al. have identified a new mechanism that regulates the activity of the TGF-beta receptor I (TbRI), which does not require interaction with the TGF-beta receptor II (TbRII), but the formation of an alternative complex between TbRI and the orphan GPR50, a member of the G-coupled receptor superfamily.

Work shows great novelty and interest, not only due to the relevance of the new mechanism described, but also because authors results suggest that this interaction might be altered in tumors that would loss the response to TGF-beta as a tumor suppressor.

Response: We thank the reviewer #1 for her/his encouraging comments and pertinent questions.

Experiments are well designed and performed and most of the conclusions are supported by the data. However, some points must be taken into consideration before publication of the manuscript:

1. It is not completely clear how the interaction between GPR50 and TbRI favors activation of the TbRI kinase capacity. One of the canonical mechanisms that activate TbRI is through phosphorylation by TbRII. Authors propose that GPR50 activation does not require TbRII. But which is the level of phosphorylation of the TbRI in the absence or presence of GPR50 expression? Is GPR50 provoking TbRI phosphorylation by alternative mechanisms that do not require TbRII? Alternatively, is the increase in Smad2 and Smad3 phosphorylation, observed after GPR50-TbRI interaction, independent of the level of TbRI phosphorylation? Authors suggest potential allosteric modifications in the TbRI as consequence of the GPR50 interaction, but they do not demonstrate how it could affect to TbRI phosphorylation. At least, authors must explore the phospho-TbRI levels by performing experiments in cells lacking TbRII, such as those used in the manuscript, after overexpressing the active form of GPR50.

Response: We agree with the reviewer that this question is very important from a mechanistical point of view. We monitored phosphorylation of T β RI at position 165 using a phosphor-Ser165 specific antibody (*Souchelnytskyi et al., EMBO J, 15(22):6231-40, 1996*) in SNU638 cells which are devoid of endogenous T β RII. Expression of T β RII and stimulation by TGF β induced T β RI-S165 phosphorylation to a similar extent as GPR50 Δ 4 and GPR50wt expression (**Figure 4c**). Phosphorylation of T β RI in the presence of GPR50 was inhibited by the T β RI kinase-specific SB431542 inhibitor suggesting the involvement of the T β RI kinase activity (**Suppl Figure 4d**). This result suggests that T β RI is autophosphorylated at Ser165 in the presence of GPR50.

2. Another alternative possibility that would explain the observed effects should be that interaction between GPR50 and TbRI would induce changes in the TbRI distribution in the cell membrane, moving it to early endosomes that would favor its interaction with SARA or other Smad2/3 scaffolds. Authors mention that they explored the TGF-beta receptor cell surface expression, but results are not shown. Furthermore, the most interesting point should be to explore potential differences in the TbRI intracellular trafficking when GPR50 is overexpressed.

Response: To address this point, we determined the subcellular localization of T β RI in the absence and presence of GPR50. Whereas T β RI did not colocalize with the early endosomal EEA1 marker proteins, substantial colocalization was observed when the wt or Δ 4 form of GPR50 was co-expressed or when cells were stimulated with TGF β (**Figure 2d**). This result supports the idea that GPR50 activates T β RI, which is then relocalized into early endosomes where it activates the Smad pathway.

When we mentioned that TGF β receptor cell surface expression is not modified by the presence of GPR50 we were referring to the number of ¹²⁵I-TGF β binding sites (see previous Suppl. Fig. 3b), which actually reflects the

number of T β RII and not T β RI. We are sorry for this confusion and deleted this notion from the revised version of the manuscript.

3. Authors mainly focus on the suppressor effects of the TGF-beta, ignoring the numerous situations where this cytokine contributes to promote tumor dissemination, migration and metastasis, particularly by its capacity to induce epithelial-mesenchymal transition. This work would require that authors present some data indicating whether or not the interaction of GPR50 with T β RI may affect to the non canonical pathways induced by TGF-beta in some tumor cells. A cancer cell line that responds to TGF-beta inducing migration/invasion, but not cytostasis, must be chosen, to explore whether this GPR50- T β RI interaction exists and which are the effects on the response to TGF-beta. It was described that phosphorylation of Smad3 is required for Snail upregulation by TGF-beta. It should be expected that GPR50, through activating T β RI, could be favoring TGF-beta pro-tumorigenic actions, too. But it is necessary to demonstrate it. Non-canonical signals involved in TGF-beta pro-tumorigenic effects, such as PI3K/AKT and ERKs, must be considered to be explored in this study too.

Response: To study the effect of GPR50 on the well-documented effect of TGF β on cell migration, we chose the 4T1 mammary carcinoma cell line that is resistant to the growth inhibitory effect of TGF β (McEarchern et al., *Int J Cancer*, 91:76-82, 2001). In the presence of GPR50 a T β RI/GPR50 complex was also formed in 4T1 cells (Suppl Figure 5c). We show here that TGF β promoted migration of 4T1 cells as expected (Figure 5d). In cells stably expressing GPR50, migration was similarly accelerated (Figure 5d) indicating that GPR50 exhibits also TGF β -like properties in respect of cell migration.

Expression of GPR50 in 4T1 cells not only increased phosphorylation of Smad3 but activated also several non-canonical TGF β signaling pathways such as p38 and AKT to a similar extent as TGF β treatment and ERK1/2 to a lesser extent (Suppl Figure 5d). Activation of these pathways by GPR50 is consistent with the promotion of cell migration by GPR50. We also observed elevated basal Snail levels, a TGF β target gene, in cells stably expressing GPR50, which was further increased by TGF β treatment (Figure 2h). This further shows the biological significance of the GPR50 effect.

Reviewer #2 (Remarks to the Author):

GPR50 is an orphan GPCR that is highly related to the two melatonin receptors, MT1 and MT2, but does not bind melatonin, and the GPR50 endogenous ligand is still unknown. It was initially shown that GPR50 interacts with and limits the activation of MT1 receptors. In a yeast-two-hybrid screen, it was found that GPR50 also acts as a signaling partner and modulator of the transcriptional co-activator TIP60, thereby regulating glucocorticoid receptor signaling. Most of these interactions occur through the large carboxyl terminal tail (C-tail) of GPR50, which functions as a scaffold for interacting partners. Using proteomics approaches, the authors have now found that GPR50 binds to T β RI through its C-tail, and that this results in increased T β RI signaling independently from T β RII. At the molecular level, evidence is presented that GPR50 enhances the basal TGF β -independent function of T β RI, increasing Smad2/3 phosphorylation likely by competing with the inhibitory binding of FKBP12 to T β RI and by stabilizing the active T β RI conformation. Evidence is also presented that GPR50 has antitumor activity in mouse models, and that GPR50 levels are associated with poor survival prognosis in selected human breast cancer studies. Overall, there is some interest in this study, which involves a large repertoire of strategies to document the interaction between T β RI and GPR50 and its functional consequences. That said, although statistically significant many of the effects appear to be subtle, and difficult to compare to the effects achieved by TGF β by acting on the T β RI-T β RII complex. While the interaction between GPR50 and T β RI can occur, this reviewer is not convinced about the biological relevance of this observation.

Response: We thank reviewer #2 for this critical assessment. We are providing now a substantial amount of additional control experiments and conditions as suggested by reviewer #2 that further strengthened our data and conclusions (see below).

What is the stoichiometry of GPR50 binding to T β RI with respect to T β RI binding to T β RII?

Response: The stoichiometry of the T β RI/GPR50 complex was estimated by cross-linking experiments followed by co-immunoprecipitation and compared to the T β RI/T β RII treated in an identical manner (Suppl Figure 1f). T β RI co-immunoprecipitated, predominantly in its monomeric form (55 kDa). Crosslinking with 0.5 mM of DSS significantly stabilized the dimeric (110 kDa) and higher-order oligomeric forms (upper right WB). Likewise, DSS crosslinking shifted the monomeric GPR50 (70 kDa) form into the dimeric form (140 kDa) and higher oligomeric forms (upper middle WB). When T β RI was co-immunoprecipitated with anti-Flag(GPR50) antibodies in the presence of DSS, T β RI was predominantly detected at >250 kDa (upper left WB). This result suggest that the molecular complex contains at least one T β RI dimer and one GPR50 dimer (120+140= 260 kDa).

When applying the same protocol to the T β RI/T β RII complex known to form a tetrameric complex composed of T β RI and T β RII dimers upon TGF β stimulation, an immunoreactive band migrating at >250 kDa was observed in coIP experiments of cross-linked samples (lower left WB). This result validates our experimental approach and strengthens the conclusion for the T β RI/GPR50 complex.

The initial interaction of GPR50 with T β RI was identified by tandem affinity purification coupled to mass spectrometry using the GPR50 Δ 4 variant in HEK293T cells.

Response: The detailed protocol and other details have been incorporated in the method section of manuscript (page 24).

What were the statistical methods used to determine the potential significance of the identification of 5 unique peptides corresponding to the T β RI in their analysis?

Response: We have implemented the “Material and Methods” in order to expose the limits set in the search engine to keep false positive rate below reasonable proportion (1%). Proteomics community sets a high confidence starting at 2 distinct peptides within these limits. Yet here, specifically to T β RI we have 5 distinct peptides respecting these limits and therefore making its identification unambiguous. Further technical details are now stated in the experimental section (page 24).

How does it compare to the other peptides identified during this study?

Response: “With a score of 214 for 5 distinct peptides, T β RI is at the 50th rank by decreasing scores, among 150 identified with at least 2 peptides (the median score value is 146, scores span from 30 to around 1600 (Na/K-ATPase subunit alpha 1: 19 distinct peptides; Calmodulin :9 distinct peptides). The list includes ubiquitous contaminants and processing-related proteins (MTR1L, Trypsin, Calmodulin).”

In Figure 1, in which the interaction is first documented, many of the panels lack appropriate controls. For example, in panel e, the control IgG IP appears to have been spliced out from the IP with anti GPR50. It is strange that the authors do not show T β RI binding to T β RII in the T β RI IP, and the bands on the total cell lysates one has to trust that they represent the appropriate proteins without the benefit of controls, for example of knock down or knock out cell lysates.

Response:

Panel e: The initial blot was separated in two parts since additional conditions (not shown) were loaded between these two lanes. To improve the readability of this panel we show now a new experiment where both lanes (IgG and anti-T β RI) are loaded side-by-side. Data are presented in the new Panel e.

Panel f: To demonstrate the specificity of the T β RI and GPR50 bands in NCI-H520 cells expressing both proteins endogenously, we performed co-IP experiments with cells treated with si-RNA molecules either directed against GPR50 or T β RI. Silencing was effective in both conditions as shown in total cell lysates (TCL) and coIP of GPR50 was substantially inhibited under these conditions as would be expected. Data are presented in the new Panel f.

We show now the presence of T β RI in the IP of T β RII in NCI-H520 cells upon treatment with TGF β as requested (Suppl Figure 1d) to complement the T β RI/GPR50 coIP.

Same of the association between the GPR50 Δ 4 and GPR50 to T β RI in BRET assays when compared to T β RI association to T β RII.

Response: We added now BRET donor saturation experiments showing the interaction between T β RI and T β RII. In the presence of TGF β , BRET₅₀ values decrease, which is consistent with the known oligomerization of both receptor subunits upon TGF β stimulation (see Figure 1i). Note, that this ligand-induced oligomerization contrasts with the constitutive behavior of the T β RI/GPR50 interaction.

The analysis of increased expression of GPR50 in response to TGF β in NCI-H520 cells is pedestrian at its best. No explanation or mechanistic information is presented, and the authors make very strong statements of a negative feedforward loop without much data supporting it.

Response: The effect of TGF β on GPR50 expression was studied in more detail at different time points and on the mRNA and protein level. A clear 2-fold increase of GPR50 expression is seen upon 24h of TGF β treatment at the mRNA and protein level (Figure 1j). This suggest that TGF β regulates GPR50 expression on the transcriptional level. The statement about the possible negative feed forward loop has been modified in the text (p. 6, paragraph 2, line 6).

What is the rationale for the selection of NCI-H520 cells for this studies, and how many other lung cancer cells express this receptors?

Response: NCI-H520 cells were selected because of their high endogenous GPR50 expression levels according to available mRNA data on Nextbio.com, a correlation engine from Illumina Inc. (San Diego, CA). Nextbio.com uses gene expression profiles derived from curated GEO DataSets and all previous published literature to provide information about expression pattern of proteins of interest. Human lung carcinoma cell lines (NCI-H520) were among the top cell lines having highest GPR50 mRNA expression levels (Geo id: 208311_at). Based on median expression levels there are a total of 63 lung cancer cell lines which express GPR50 endogenously. Among them top five have high expression (NCI-H520, NCI-h460, ChaGo-K-1, NCI-H1781) and the remaining 58 cell lines have moderate to low expression (NCI-H1563, NCI-H2107, NCI-H748, NCI-H1606 etc...).

In Figure 2 a, why they need to over expressed Smad3? In Figure 2 b, one would have expected that the authors would have used at least 2 siRNAs, and to use TGF β stimulation and T β RI knock down as controls. The latter is a concern thought the study, as it is hard to judge whether many of the subtle changes caused by GPR50 expression or knock down/out are meaningful (in addition to statistically significant) with respect to those caused by canonical activation of the pathway. This is clear, for example, in Figure 2 e, in which nuclear pSmad2/3 is barely affected by GPR50 Δ 4 expression, and almost nothing by GPR50. Same for reporter activation (Figure 2 f and g).

Response:

Panel a: In this panel, we show data in HEK293 cells overexpressing Smad3 since endogenous Smad3 levels are too low to see even the effect of TGF β stimulation (our positive control condition). For Smad2 we were able to detect phosphorylation at endogenous levels in cells either treated with TGF β or transfected with GPR50 (see below).

However, this was only observed on immunoprecipitated Smad proteins and not in cell lysates. We therefore decided to overexpress Smad proteins in the HEK293 cell system. The physiological relevance of the effect of GPR50 on Smad phosphorylation is indicated by the lower Smad phosphorylation levels in the cortex and hypothalamus of GPR50KO mice as shown in Figures 2c-d. Similarly silencing of GPR50 showed also an effect on endogenous Smad2 phosphorylation levels in NCI-H520 cells with an amplitude that was comparable to that obtained by T β RI silencing (Figure 2b). Furthermore, GPR50 expression significantly increased mRNA and protein levels of the endogenous TGF β target gene Snail. Altogether, these results indicate that GPR50 has a biologically relevant effect on the TGF β / Smad pathway.

Panel b: We performed a new set of experiments including the requested control conditions: stimulation by TGF β was added as a positive control, a second siRNA molecule directed against GPR50 and siRNA molecules directed against T β RI. Both siRNA molecules effectively downregulated their respective targets and diminished basal Smad2 phosphorylation as predicted. The initial set of experiments (WB and quantification) using the first siRNA molecule directed against GPR50 was transferred to Supp Figure 2c. New data are shown in the new Figure 2b.

Previous panel e: Following the reviewers comments, we optimized the experimental conditions and redesigned the experimental setting by including TGF β stimulation (0.5ng/mL) as a positive control alone as well as in the presence of the T β RI kinase-specific SB431542 inhibitor (10 μ M) as negative control. Results show that GPR50 promotes nuclear Smad2/3 translocation to a similar extent as TGF β treatment indicating that the magnitude of the effect of GPR50 is comparable to that of the established TGF β effect. Results are shown in the new Figure 2f.

Previous panels f, g: To better compare the amplitude of the effect of GPR50 expression on ARE-Luc and CAGA-Luc reporter gene expression, we are showing now the effect of TGF β stimulation (0.5ng/ml for 8h) in Mock transfected cells. The results show that GPR50 increases ARE-Luc activity to a similar extent than TGF β and CAGA-Luc to a lesser extent. Results are shown in the new Figure 2g and Supp Figure 2f.

In Figure 3 a, it is hard to tell whether GPR50 really competes for FKBP12 binding to T β RI, and in every case this competition is observed after large overexpression. The data should be compared to the effect of TGF β by acting on T β RI/II. Same for the normalized BRET association and T β RI kinase activation assays, which when

normalized to control lysate is quite limited, making it difficult to assess its potential biological relevance.

Response: Following the suggestions of the reviewer, we performed a new set of experiments for the dissociation of FKBP12 from T β RI including a positive control condition consisting of TGF β -induced FKBP12 dissociation from the T β RI/T β RII complex. As shown in the new Figure 3a, GPR50 dissociated the T β RI/FKBP12 complex to a similar extent (approximately 50%) as T β RII upon TGF β stimulation indicating the biological relevance of the effect of GPR50.

BRET experiments in the presence of FKBP12 were repeated by comparing the inhibitory effect of FKBP12 on the T β RI/GPR50 interaction and on the TGF β -induced T β RI/T β RII interaction. A decrease of 40% and 20%, respectively, was observed indicating that the effect of GPR50 is at least as strong as that of T β RII. Results are now shown in the new Figure 3d.

Concerning the T β RI kinase activity previously shown in panel h, we agree that the quantitative aspect of this experiment is limited when using cell lysates and is difficult to be compared to the T β RII promoted T β RI activation in the presence of TGF β . Given this limitation, we propose to transfer this panel into Suppl. Figure 4c and scale down its interpretation.

Taken together, the control conditions of the known TGF β -T β RI/T β RII pathway in coIP and BRET experiments indicate that the effect of GPR50 on the T β RI/FKBP12 interaction is of similar amplitude as that on the T β RI/T β RII interaction induced by TGF β . These in vitro studies are fully compatible with the observed increase of the T β RI/FKBP12 interaction as determined in lysates of brain and lung of GPR50 KO mice expressing both interacting partners at endogenous levels (Figures 3b-c) further supporting the biological relevance of the GPR50 effect.

In Figure 5 c, the increased proliferation rate promoted by knockdown of GPR50 is greater than that caused by T β RI, which when aligned with the comments above, suggest that at least part of GPR50 biological functions are independent from T β RI. The usefulness of the over expression experiments in Figure 5 b is questionable. It is somehow surprising that the authors have used mock transfected cells for this and other tumor studies and biological experiments. Were mock transfected cells subjected to similar selection than GPR50/ GPR50 Δ 4 stably expressing cells (G418 resistant)? This may altered cellular behavior, skewing the cell populations used for the studies.

Response: We agree that experiments performed under protein overexpression conditions have to be carefully interpreted as it is difficult to exclude any artificial effect due to the overexpression. That is the reason why we also performed in vitro GPR50 silencing (siRNA) and in vivo GPR50 knockout studies (Figures 6a-d). Combining both approaches overexpression and silencing is complementary. Our results obtained with both approaches are consistent and support each other.

We used indeed mock transfected cells carrying the empty vector conferring resistance to G418. Both mock and GPR50 cells were submitted to the same selection process to avoid any caveats due to differences in the selection process.

In Figure 6, it is unclear what is the meaning of “time after tumor onset”, as these mice develop tumors quite late in life, when around 200 days old, and there was no difference in tumor free survival between wt and GPR50 KO mice (sup Fig 6 d). Were these control littermates?

Response: Time after tumor onset refers to the age of the animals when tumors were first detected in each animal (frequency of detection was once a week). This time is then taken as time “0” for the calculation of the “Time after tumor onset for each animal. This is now mentioned in the “Methods” section (page 25, last paragraph). Based on this calculation, GPR50KO mice died faster than WT mice because tumors developed faster.

With thousands of breast cancer patients studied in TCGA and other efforts by RNAseq, it is unclear why the authors cherry picked some studies using gene array technologies that were conducted prior to this new database. Indeed, Prognoscan.org was last updated in 2012, before TCGA, and datasets are quite old as well.

Response: We took the reviewer's valuable suggestion into account and analyzed the prognostic value of GPR50 using the Kaplan-Meier Plotter (<http://kmplot.com/analysis>). This large breast cancer database combines information from TCGA, GEO and EGA (European Genome-phenome Archive) and has been recently updated (on the 10/13/2016) to include now 5,143 samples. This independent and large data set allowed us to confirm the prognostic impact of GPR50 expression on relapse-free survival in breast cancer patients. Low GPR50 expression is associated with poor survival prognosis even after accounting for molecular breast cancer subtypes, indicating that GPR50 is an independent prognostic marker in breast cancer. Results are shown in new figure 6f-g.

Few additional comments:

Is GPR50 expressed in immune cells? This may affect tumor development independent from TGF β signaling.

Response: GPR50 expression has not been observed in primary lymphoid tissues of mice (Regard, et al., 2008, 135:561-71, Cell), and humans (Fagerberg et al, 2014, 13:397-406, Mol Cell Proteomics), which suggest unlikely expression in immune cells.

Are G proteins involved or all effects are G protein (or arrestin) independent?

Response: Control experiments have been performed and no effect of Gi and Gq protein activation or β -arrestin silencing has been observed (see Suppl Figure 2j).

Given the rest of the available information regarding the proposed central role of the association of the GPR50 C-tail with T β RI, the authors may want to explain their interpretation of the data that cytosolic C-tail (GPR50Cter) alone or fused to the 7TM domain of the melatonin receptor MT2 (MT2-Cter GPR50) are not able to replicate the limited spontaneous activation of the Smad pathway.

Response: One likely hypothesis to explain this observation is that the C-tail of GPR50 has to be in close proximity to T β RI, i.e. in a molecular complex, a condition that cannot be replicated with the soluble GPR50 C-tail nor when fused to the MT₂ receptor that does not interact with T β RI. This aspect has now been discussed on page 14.

Mice KO for T β RI exhibit a remarkable phenotype, while GPR50 KO mice are viable and exhibit subtle phenotypes. This seems to represent a disconnect that has not been addressed and/or discussed.

Response: T β RI knockout mice die at embryonic stages and cannot survive after E10.5 because of severe vascular defects (*Larsson et al., 2001, 20:1663-73, EMBO J*). However, GPR50 expression during development has revealed that it starts expressing at E13 and peaks at E18 in several brain regions (*Grunewald et al, 2012; 42(4):363-71 ACS Chem Neurosci*). Expression of GPR50 in the vascular system during development is unknown. More generally, differential expression in terms of developmental stages, cell types and tissues are likely to exist between T β RI and GPR50 are likely to exist and maybe at the origin of the embryonic lethality of T β RI knockout mice as compared to GPR50 knockout mice. This notion is now added in the discussion on page 18.

REVIEWERS' COMMENTS:

Reviewer #1 (Remarks to the Author):

Authors have correctly addressed all the concerns. Results now are more robust and confirm the hypotheses. The work shows great interest in the area of TGF-beta signaling.

Reviewer #2 (Remarks to the Author):

The authors have addressed most major concerns, and the new information added has strengthened the present manuscript. The authors should be commended for their efforts.